

# The clay mineralogy rather that the clay content determines radiocaesium adsorption in soils

Margot Vanheukelom[1,2], Nina Haenen[1,2], Talal Almahayni[1], Lieve Sweeck[1] (deceased), Nancy Weyns[3], May Van Hees[1], Erik Smolders[2]

[1]Biosphere Impact Studies, Belgian Nuclear Research Centre (SCK CEN), Boeretang 200, Mol, 2400, Belgium
[2]Division of Soil and Water Management, KU Leuven, Kasteelpark Arenberg 20, Leuven, 3001, Belgium
[3]Division of Geology, KU Leuven Celestijnenlaan 200e – box 2411, Leuven, 3001, Belgium

*Correspondence to*: Margot Vanheukelom (margot.vanheukelom@kuleuven.be)

**Abstract.** The transfer of radiocaesium ($^{137}$Cs) from soil to crops is the main long-term radiation risk after nuclear accidents.

The prevailing concept is that $^{137}$Cs sorption in soil, and hence its bioavailability, is controlled by soil clay content (0–2 µm). This study tested this assumption using 24 soils collected worldwide. The Radiocaesium Interception Potential (RIP), i.e., $^{137}$Cs adsorption, was measured for the bulk soils and for their clay and silt fractions. The RIP varied by factor 438 among soils and was *unrelated* to its clay content ($p > 0.05$). The RIP in the clay fractions was lowest for young volcanic soils with allophane and mica, and for highly weathered tropical soils with kaolinite. In contrast, about two order of magnitude higher

RIP values were found in intermediate-weathered temperate soils dominated by illite. Soil RIP was, hence, related to soil illite content ($R^2 = 0.50$; $p < 0.001$). Significant fraction of soil RIP originated from clay minerals embedded in the silt fraction. The sum of RIP in clay and silt fractions overestimated the soil RIP by, on average, factor of 2, indicating that isolation of clay opens selective $^{137}$Cs sorption sites inaccessible in intact soils. Soil mineralogy, not just clay content, governs soil RIP. The validity of existing $^{137}$Cs bioavailability models require recalibration for its use on a global scale.





**Graphical abstract**

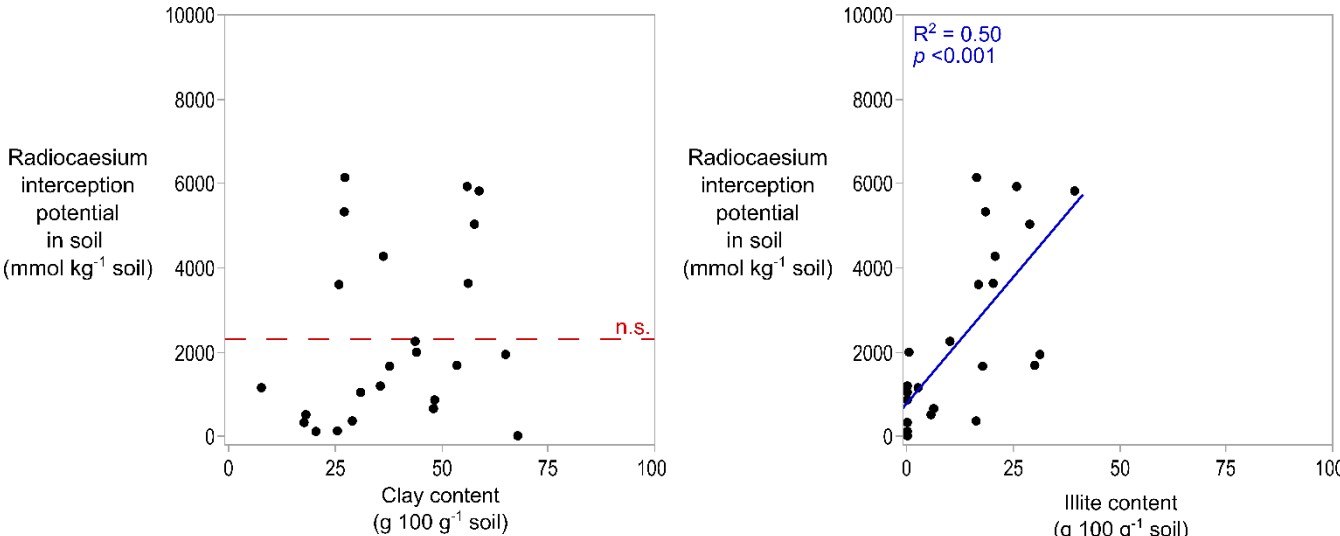

**1 Introduction**

Global interest in nuclear energy requires environmental impact analyses that rely on models predicting the transfer of radiocaesium from soil to plants. There are concerns about radiocaesium ($^{137}$Cs) due to the long half-life (30 years) and

biochemical similarity to potassium (K), a major plant nutrient, which allows it to transfer from soil into the food chain. Following nuclear incidents, such as Chornobyl and Fukushima, models were developed to predict $^{137}$Cs soil-to-plant transfer in affected regions (Absalom et al., 1999, 2001; Tarsitano et al., 2011; Uematsu et al., 2016). These models account for $^{137}$Cs adsorption in soil and the effects of competing ions, mainly K, that control its absorption and translocation to the edible fraction of the crop. Radiocaesium, in trace concentrations, is selectively adsorbed into wedge-shaped edges of micaceous clay minerals

in soils formed by weathering (Fanning et al., 1989; Sawhney, 1972), called frayed edge sites (FES). The radiocaesium interception potential (RIP), reflecting selectivity for $^{137}$Cs$^+$ and the capacity of adsorption sites, can be easily quantified by laboratory tests (Cremers et al., 1988; Wauters et al., 1996b). The RIP in soil depends on both the content of FES-bearing clay minerals and the amount of FES within these minerals, which varies with clay mineral type and weathering stage (Maes et al., 1999a, b). Thus, the bioavailability of $^{137}$Cs in soils is influenced by soil mineralogy, which depends on the parent material

and the weathering stage.

However, the above-mentioned soil-to-plant transfer models typically assume a uniform mineral composition, which does not accurately reflect the diversity in soil mineralogy on a global scale. These models assume that the RIP in the soil is related to the clay content, and the RIP in the soil is derived by multiplying soil clay content, i.e., 0–2 µm fraction, with the RIP of the clay fraction in soil (RIP$_{clay}$). In so doing, the models implicitly assume that (1) clay minerals with FES, commonly found in

the clay fraction, are absent in other soil texture fractions, such as silt or sand, and (2) variations in the mineral composition of



the clay fraction itself do not significantly affect the overall RIP of the soil. Despite these simplifications, these models have shown a high predictive power, even across soils from contrasting locations, weathering stages, and soil mineralogy (Vanheukelom et al., 2024). However, a proper fit of the [137]Cs bioavailability data does not imply that the model assumptions are valid because compensating mechanisms, e.g., incorrect estimates of available K, occur, as recently discussed (Vanheukelom et al., 2024). In a recent study where different clay minerals were mixed with sand, it was shown that the RIP and [137]Cs transfer to grass varied over two orders of magnitude at equal clay content (Vanheukelom et al., 2023). A follow-up study with natural soil samples collected globally showed that the soil RIP divided by its clay content (RIP/clay) varied three orders of magnitude (Vanheukelom et al., 2024). Lowest RIP/clay ratios were found in soils dominated by young clays, such as allophane and mica (740 mmol kg$^{-1}$ clay), and in soils with highly weathered clays, such as kaolinite (320 mmol kg$^{-1}$ clay). The largest ratios were found in soils of temperate regions in which illite dominates (26,000 mmol kg$^{-1}$ clay). In that study, the RIP and soil mineralogy were determined only on the bulk soils, but the mineralogy and RIP of the clay and silt fractions were not determined.

This study aims to verify or falsify the main premise of radiocaesium soil-to-plant transfer models, i.e., that soil RIP is primarily controlled by its clay content and that the RIP of the clay fraction is fairly constant. Specifically, we measured RIPs in soils and in their corresponding isolated texture fraction (0–2 µm and 2–50 µm) and measured mineralogy by X-ray diffraction (XRD) in both fractions. We selected soils of two weathering sequences from Kenya ($N = 8$), the Philippines ($N = 6$), and soils of various parent materials and weathering stages sampled around the world ($N = 10$). The 0–2 µm and 2–50 µm fractions were isolated using ultrasonic disaggregation and Na$^{+}$-resin without removing any cementing soil constituents such as organic matter or iron minerals, reflecting their condition in the intact soil.

## 2 Materials and methods

### 2.1 Soil collection

Soil sequences sampled in Kenya ($N = 8$) and the Philippines ($N = 6$) represented soils at various weathering stages. In addition, soils were sampled at multiple locations worldwide ($N = 10$), representing soils from various parent materials. The geology of the parent materials was derived from a global map (onegeology.org). Soils from Japan (Miyakonojo) were derived from volcanic rocks from non-alkali pyroclastic flows and (Ogata; Yoshiki) from sedimentary rocks of the Late Pleistocene-Holocene. Soils from the Philippines were derived from felsic volcanic rocks of the Quaternary period. Soil from Spain (Isla Mayor) was derived from alluvial deposits of the Holocene epoch. Soils from Austria (Gross Enzersdorf) and United States (Sidney) were derived from Cenozoic sedimentary rocks, and in Italy (Aliminusa) this deposit was turbidite. Soils from Kenya (Mount Elgon) were derived from Cenozoic volcanic rocks and (Endebess; Kitale) from Neoproterozoic metamorphic rocks, including andesite and dacite. Soil from China (Changchun) was derived from Paleozoic felsic plutonic rocks. Soil from Madagascar (Behenjy) was derived from Neoarchean plutonic and metamorphic rocks. The intact soil samples, noted as bulk



soils, were characterized in a previous [137]Cs pot experiment with grass (Vanheukelom et al., 2024), except for two soils from Kenya (Mount Elgon 3 and Kitale 3), which were characterized following the same procedures.

## 2.2 Clay and silt isolation from soil

The texture fractions were isolated from soils by mechanical disaggregation and dispersion using a sodium cation resin (Na[+]-resin) (Rouiller et al., 1972). This method was preferred over the widely used standardized pipet method with removal or organic matter or other cementing materials (ISO, 1998) as it is assumed to be a less biased method and appropriate for andic (Delvaux et al., 1989) and ferralitic (Bartoli et al., 1991) soils. Air-dried bulk soils were milled and sieved (2 mm). Air-dried bulk soil weights were corrected for moisture (60 °C). Deionized water (solid : liquid of 1 g soil : 20 mL water) was added to

the soils and shaken end-over-end (16 h). Sand fractions were separated by wet sieving (50 µm; certified DIN ISO 3310-1; VWR®) and were repeatedly ultrasonically treated (15 min at 90 J min[-1]; Misonix Sonicator® XL2020) until the sand grains were clear. The <50 µm fractions were flocculated by adding drops of saturated $CaCl_2$ so that the excess water could be removed by siphoning. The Na[+]-resin (>500 µm; Amberlite® IR-120 (H); Supelco®) was added to <50 µm fractions (1 g soil : 10 mL resin) and were shaken end-over-end (16 h). The resin was removed by wet sieving (500 µm superimposing 50 µm)

and washed with deionized water until the filtrate was clear. The <50 µm fractions in deionized water (500 mL) in graduated cylinders were placed in a water bath (20.0 ± 0.5 °C). The <50 µm was plunged, and the 0–2 µm fraction (clay fraction hereafter) in the suspension was siphoned off 20 cm below the surface using an upward pointed tube connected to a pump after 15 h 28 min 18 sec. Plunging and siphoning were repeated (3–5 times) until the suspension was clear. The remaining 2–50 µm fraction (silt fraction hereafter) was recovered. All fractions were dried in the oven (60 °C; 48 h). Fractions were cooled to

room temperature in a desiccator and weighted analytically (accuracy 0.0001 g; OHAUS Adventurer™ AR2140). The particle size distributions were compared between the Na[+]-resin method and the pipet method using reference soils from proficiency tests (Table A1). The clay contents for three of the four soils matched very well for both methods, but for one soil (from a temperate region), the resin method identified twice as much clay as the pipet method.

## 2.3 X-ray diffraction (XRD)

The mineralogy of clay and silt fractions was analyzed by XRD. The mineralogy of bulk soils was quantified, and the soil weathering index was calculated, which was previously reported (Vanheukelom et al., 2024; Table B1 in this study). The clay and silt fractions were prepared as oriented slides to enhance the basal reflections (Moore and Reynolds, 1997) in air-dried Ca-form and solvated ethylene glycol form. The fractions in deionized water (5 mL) were ultrasonically disaggregated (1 min at 420 J min[-1]; Sonics Vibra-cell™ VCX-130) and put in dialysis membrane bags (6–8 kDa; Spectra/Por®). These were immersed

in 1 mol L[-1] $CaCl_2$ solution (250 mL; 16 h). Excess electrolytes were removed by replacing the outer solution with deionized water and shaking until the conductivity of the outer solution matched that of deionized water. The fractions were recovered from the dialysis bags and dried in the oven (60 °C; 16 h). Subsamples of clay and silt fractions were taken to make smear slides on glass plates for XRD measurement. Dried fractions were deposited on a glass plate (10 mg cm[-2]), and two drops of





deionized water were added to form a paste that was smeared until a uniform, thick film with a smooth surface was obtained.

The Ca-saturated smear slides were air-dried and measured by XRD. The smear slides were placed in a desiccator filled with ethylene glycol and in the oven (60 °C; 16 h). The smear slides solvated with ethylene glycol were measured by XRD. The XRD device (Bruker D8 Advance) in reflection geometry with Cu-K$_\alpha$ radiation with Ni-K$_\beta$ filter had a 435 mm goniometer radius, rotating sample holder (15 rpm), divergence slit (0.25°), fixed anti-scatter slits (5.69 mm), soller slits (2.5°), detector slit (11.55 mm), a multistrip detector (Bruker Lynxeye) scanning at 40 kV and 30 mA, ranging over 2–47 °2θ with 0.015 °2θ

step size and 0.5 s counting time. Mineralogies of clay fractions were quantified, and silt fractions were identified by Sybilla software (© Chevron ETC) that used the multispecimen method by fitting the model on the XRD record in air-dried Ca-form and in ethylene glycol solvated states (Zeelmaekers, 2011). A constant function removed the background, patterns were shifted to match kaolinite mineral peaks, and the step size was increased (0.05 °2θ) to facilitate calculations, which was acceptable given the wide clay peaks. The results of quantified phases were accepted if the model fit in the ethylene glycol state matched

the model in the air-dry state, and the average of quantified phases in two states was taken (Table 2; Table C1).

## 2.4 Radiocaesium interception potential (RIP)

The RIP of clay and silt fractions was analyzed by exchanging $^{137}$Cs with K-cations at selective adsorption sites and blocking non-selective sites with Ca-cations (Wauters et al., 1996a). The RIP of bulk soils were previously reported (Vanheukelom et al., 2024; Table B2 in this study). Subsamples of clay and silt fractions (0.2–0.4 g) were put in dialysis membrane bags (6–8

kDa; Spectra/Por®). A solution of 0.5 mmol K L$^{-1}$ and 100 mmol Ca L$^{-1}$ was added to membrane bags (10 mL) and in pots (150 mL). Membrane bags were immersed in the pots and shaken end-over-end (24 h). The outer K-Ca solution (150 mL) was renewed and shaken (24 h). Next, the outer solution was renewed with carrier-free 200 kBq $^{137}$Cs mL$^{-1}$ with K-Ca solution and shaken (24 h). The $^{137}$Cs concentration in the initial and final outer solution was measured with a gamma counter (Perkin Elmer 1480 WIZARD 3"). The RIP was calculated by multiplying the measured $^{137}$Cs solid-liquid distribution coefficient by 0.5

mmol K L$^{-1}$ in solution.

## 2.5 Statistical analyses

JMP® Pro software (Version 17.0.0 SAS Institute Inc., Cary, NC, 1989–2024) was used. Pearson correlation coefficients were calculated with a pairwise method on the means ($n = 1$ or $n = 3$) or log-transformed means if skewness was closer to 0 after transformation (Table D1). Linear regression was used ($p < 0.05$) to obtain relations describing RIP with selected explanatory

variables.





# 3 Results

## 3.1 Properties of bulk soil and fractions

The grouping of the 24 soil samples by weathering stages showed that the youngest soils had the highest carbon contents (5.2 g 100 g$^{-1}$) and the most weathered soils had finer textures (32 g 100 g$^{-1}$), while soils in the intermediate stage of weathering

had the highest pH and CEC (5.9 and 23 cmol$_c$ kg$^{-1}$; Table 1).

**Table 1: Properties of bulk soils as weathering index (WI) increased (detailed in Table B1 and Table B2). The number of soils per group (*N*), minimum (min), maximum (max), and geometric mean (GM) for each group are given.**

| soil group[a] | *N* | weathering index | | | <2 µm | | | 50–2000 µm | | | organic C | | |
|---|---|---|---|---|---|---|---|---|---|---|---|---|---|
| | | | | | | | g 100 g$^{-1}$ | | | | | | 140 |
| | | min | max | **GM** | min | max | **GM** | min | max | **GM** | min | max | **GM** |
| *young* | 3 | 2.5 | 4.6 | **3.5** | 20 | 31 | **26** | 9 | 38 | **22** | 1.8 | 14 | **5.2** |
| *intermediate* | 13 | 4.8 | 6.9 | **5.6** | 8 | 65 | **33** | 3 | 73 | **19** | 0.8 | 8.4 | **1.9** |
| *weathered* | 7 | 7.1 | 9.2 | **8.0** | 25 | 68 | **43** | 20 | 68 | **32** | 0.6 | 2.9 | **1.7** |

[a]soils are grouped as young (WI = 2.5–4.7), intermediate (WI = 4.7–7.0), or weathered (WI = 7.0–9.2) based on X-ray diffraction
mineralogy measured on bulk soil

Table 1: *Continued*.

| soil group[a] | *N* | pH | | | CEC | | | RIP$_{soil}$ | | |
|---|---|---|---|---|---|---|---|---|---|---|
| | | | | | cmol$_c$ kg$^{-1}$ | | | mmol kg$^{-1}$ | | |
| | | min | max | **GM** | min | max | **GM** | min | max | **GM** |
| *young* | 3 | 4.1 | 5.9 | **4.9** | 5.1 | 24 | **9.2** | 120 | 1,000 | **350** |
| *intermediate* | 13 | 4.1 | 7.8 | **5.9** | 5.4 | 50 | **23** | 330 | 6,100 | **2,400** |
| *weathered* | 7 | 3.8 | 4.8 | **4.3** | 2.3 | 28 | **11** | 14 | 3,600 | **570** |

## 3.2 Mineralogy of the soil fractions

Not all mineral phases could be quantified (Fig. C3 to Fig. C26), so the mineralogy of the fractions was only indicative. The Sybilla software library only included mineral phases of illite, expanding vermiculite or smectite, kaolinite, and chlorite
phyllosilicates, and interstratified combinations of these. Other phases, such as allophane, mica, and feldspars, could not be quantified, so mineralogies of the clay fractions were proportions of a limited selection of the aforementioned phyllosilicates (Table 2; Table C1).

It was assumed that the soil sequences were derived from the same parent material, i.e., with similar mineral composition, so similar mineral phases were identified within the sequence. The sequence of soils from Kenya had enrichment of kaolinite and
illite in the clay fraction with increasing weathering (Fig. C1). The youngest soil at the highest sampling point on the mountain (Mount Elgon 1) did not have clear peaks of phyllosilicates that could be distinguished from the background noise in the XRD pattern (Fig. C4). However, the right-tailed broad peak around 4.4–4.5 Å of hkl reflections indicated the presence of disordered 1:1 phyllosilicates, such as halloysite or poorly crystallized, fine-sized kaolinite. Wider humps around 3.52 Å and 2.50 Å





confirmed the presence of allophane, previously determined by oxalic acid extraction (3 g allophane 100 g⁻¹ bulk soil;
Vanheukelom et al., 2024) In more weathered soils on the mountain slope (Fig. C18), the kaolinite peak was clear around
7.20–7.50 Å. This broad peak indicated a mixed-layer mineral phase with kaolinite. A small peak around 10.1 Å indicated
illite mineral phases. In a moist valley soil (Endebess; Fig. C20), a broad peak at 15.3 Å that shifted to 16.3 Å after ethylene
glycol treatment indicated swelling phyllosilicates, which was to be expected in a Vertisol (Table B2). In the most weathered
upland soils (Kitale 2 and 3; Fig. C23 and Fig. C24). the kaolinite and illite peaks were largest (Fig. C1).

The sequence of soils from the Philippines showed enrichment of kaolinite in the clay fractions with increasing weathering
(Fig. C2). In the youngest soil (Pagsanjan 1; Fig. C5), a peak at 14.8 Å shifted to 16.6 Å by ethylene glycol indicating the
presence of expanding phyllosilicates. In one soil (Pagsanjan 5; Fig. C22), a small peak at 11.3 Å suggested the presence of
illite, but it was barely detectable in the XRD pattern after ethylene glycol treatment. The most weathered soil of the sequence
(Cavinti; Fig. C26) showed the largest kaolinite peaks at 7.28 Å and 3.59 Å.

The variability in mineralogy between young and highly weathered soils was even more pronounced in the other soils that
differed in parent material and weathering stage. The youngest soil from Miyakonojo (Japan), derived from volcanic ash,
contained allophane characterized by wide humps around 3.50 Å and 2.50 Å in the clay fraction (Fig. C3) and what was
previously confirmed by oxalic acid extraction (11 g allophane 100 g⁻¹ bulk soil; Vanheukelom et al., 2024). But otherwise,
no clear peaks of phyllosilicates could be identified in the volcanic soil. Soils in the intermediate stage of weathering, such as

Rots (France; Fig. C12) and Isla Mayor (Spain; Fig. C15) had mainly peaks of illite at 10 Å, 5 Å, and 3.33 Å. In the same
group of soils with intermediate weathering stage, expanding phyllosilicates with shifting peaks after ethylene glycol treatment
were very clear in Ogata (Japan) with 15.3 Å peak shifting to 16.7 Å (Fig. C10) and in Changchun (China) with 14.3 Å peak
shifting to 17.5 Å (Fig. C13). These soils with intermediate stage of weathering also contained, to a lesser extent, kaolinite
with peaks at 7.1–7.2 Å. In some soils, such as Yoshiki, Japan, mica was identified in the silt fraction by peaks around 10 Å,

5 Å, and 2.55 Å (Fig. C27). The weathered soil from Behenjy (Madagascar) had peaks of kaolinite at 7.16 Å, 4.37 Å, and 3.57
Å (Fig. C25).





**Table 2: Mineralogy of the clay fraction and RIPs of the clay and silt fractions of soils grouped with increasing weathering index (detailed in Table C1). I = illite; S = expanding phyllosilicate; C = chlorite; K = kaolinite. Note: Mineral phases quantified are limited to illite, vermiculite, smectite, kaolinite, chlorite, and their mixed layers; other phases (e.g., allophane, mica, feldspars) were not included in the analysis.**

| location | weathering index | mineralogy clay fraction | | | | RIP | |
| | | I | S | C | K | <2 µm | 2–50 µm |
| city, country | | ——— g 100 g$^{-1}$ ——— | | | | ——mmol kg$^{-1}$ —— | |
| *young* | | | | | | | |
| Miyakonojo, Japan | 2.5 | <0.5 | <0.5 | <0.5 | <0.5 | 370 | 230 |
| Mount Elgon, Kenya (1) | 3.7 | 56 | 16 | <0.5 | 28 | 1,200 | 1,100 |
| Pagsanjan, Philippines (1) | 4.6 | <0.5 | 35 | <0.5 | 65 | 2,100 | 420 |
| *intermediate* | | | | | | | |
| Sidney, United States | 4.8 | 60 | 33 | <0.5 | 7 | 14,000 | 2,900 |
| Mount Elgon, Kenya (2) | 5.1 | 56 | 15 | <0.5 | 29 | 3,900 | 2,800 |
| Pagsanjan, Philippines (2) | 5.1 | <0.5 | 42 | <0.5 | 58 | 1,200 | 500 |
| Gross Enzersdorf, Austria | 5.2 | 65 | 31 | 1 | 3 | 16,000 | 1,500 |
| Ogata, Japan | 5.2 | 1 | 66 | <0.5 | 33 | 6,500 | 3,800 |
| Mount Elgon, Kenya (3) | 5.4 | 48 | 18 | <0.5 | 34 | 5,300 | 2,100 |
| Rots, France | 5.5 | 68 | 23 | <0.5 | 9 | 16,000 | 790 |
| Changchun, China | 5.8 | 57 | 27 | <0.5 | 16 | 11,000 | 2,300 |
| Pagsanjan, Philippines (3) | 6.0 | <0.5 | 31 | <0.5 | 69 | 2,300 | 1,200 |
| Isla Mayor, Spain | 6.0 | 67 | 18 | 6 | 9 | 14,000 | 2,400 |
| Yoshiki, Japan | 6.0 | 34 | 14 | <0.5 | 52 | 7,500 | 5,600 |
| Kitale, Kenya (1) | 6.1 | 31 | 10 | <0.5 | 59 | 3,800 | 440 |
| Mount Elgon, Kenya (4) | 6.6 | 50 | 18 | <0.5 | 32 | 7,100 | 4,000 |
| Aliminusa, Italy | 6.7 | 46 | 26 | <0.5 | 28 | 19,000 | 1,500 |
| *weathered* | | | | | | | |
| Endebess, Kenya | 7.1 | 47 | 20 | <0.5 | 33 | 5,800 | 720 |
| Pagsanjan, Philippines (4) | 7.5 | <0.5 | 36 | <0.5 | 64 | 4,900 | 2,000 |
| Pagsanjan, Philippines (5) | 7.6 | 13 | 18 | <0.5 | 69 | 12,000 | 1,200 |
| Kitale, Kenya (2) | 7.9 | 36 | 10 | <0.5 | 54 | 6,700 | 3,700 |
| Kitale, Kenya (3) | 7.9 | 23 | 6 | <0.5 | 71 | 4,900 | 3,000 |
| Behenjy, Madagascar | 9.0 | <0.5 | <0.5 | 53 | 47 | 100 | 68 |
| Cavinti, Philippines | 9.2 | <0.5 | 12 | <0.5 | 88 | 1,400 | 34 |



### 3.3 $^{137}$Cs adsorption in bulk soil and fractions

The RIP of bulk soils was lowest in the youngest (350 mmol kg$^{-1}$) and most weathered soils (570 mmol kg$^{-1}$). Soil RIP was not correlated to the texture, organic C, or CEC (Table D1) but was positively correlated with pH ($r = 0.66$; $p < 0.001$), possibly

because pH acts as a proxy for other factors influencing RIP, as pH has no direct effect on $^{137}$Cs sorption in soils (Wauters et al., 1994). Lower pH often characterizes highly weathered soils with fewer FES-bearing clay minerals. Notably, RIP measured in bulk soils was *not* explained by the clay content ($p > 0.05$) (Fig. 1).

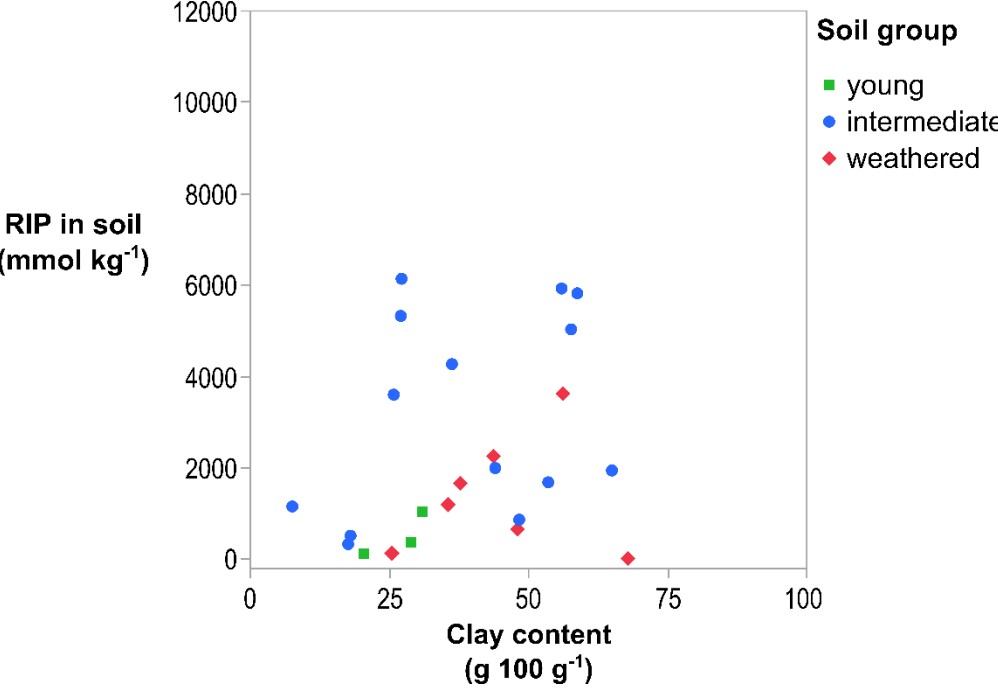

**Figure 1: The RIP in bulk soil is not linearly related to its clay content of the soil ($p > 0.05$).**

The RIP was larger in the clay than in the silt fraction for all soils (Table 2). The RIP in the clay fraction varied with soil weathering stages (Fig. 2). The youngest soils had a RIP corresponding to that measured in mica (300–740 mmol kg$^{-1}$;.Eguchi et al., 2015; De Preter, 1990) Intermediate-weathered soils had a highly variable RIP, and soils dominated by illite had a RIP similar to that measured in pure illite (12,600–16,000 mmol kg$^{-1}$;.de Koning et al., 2007; Wauters et al., 1994) Weathered soils had a low RIP when illite was absent, corresponding to the RIP measured in kaolinite (6–310 mmol kg$^{-1}$;.Nakao et al., 2008;

Ogasawara et al., 2013)

The RIP in the clay fraction can be explained by its mineralogy and, overall, the RIP in the clay fraction was positively correlated with its illite content ($R^2 = 0.43$; $p < 0.001$; Fig. 3). Young soils had a low RIP (<3,000 mmol kg$^{-1}$) in the clay fraction with low illite content (<0.5 g 100 g$^{-1}$), except for a soil from Mount Elgon (Kenya 1; illite >50 g 100 g$^{-1}$), but the illite content was likely overestimated by XRD quantification by not taking into account other mineral phases, such as quartz, hematite, and



allophane. Intermediate-weathered soils, dominated by illite (>30 g 100 g$^{-1}$), had the highest RIP (up to 19,000 mmol kg$^{-1}$). Except for Mount Elgon (Kenya 2), which, despite its high illite content (>50 g 100 g$^{-1}$), had a low RIP for reasons explained above. Other intermediate-weathered soils, dominated by expanding phyllosilicates (i.e., high-charge smectite and regular smectite >30 g 100 g$^{-1}$), had lower RIP (<10,000 mmol kg$^{-1}$). More weathered soils, dominated by kaolinite (up to 88 g 100 g$^{-1}$), had low RIPs (<10,000 mmol kg$^{-1}$). However, a weathered soil from the Philippines sequence (Pagsanjan 5) with over factor

5 higher RIP (12,000 mmol kg$^{-1}$) compared with other soils from that sequence (geometric mean (GM) = 2,100 mmol kg$^{-1}$; $N$ = 5) had illite (13 g 100 g$^{-1}$) while it was not detectable in other soils. The lowest RIP (100 mmol kg$^{-1}$) in the clay fraction was measured in a highly weathered soil from Behenjy (Madagascar) in which illite or expanding phyllosilicates were not detectable (<0.5 g 100 g$^{-1}$).

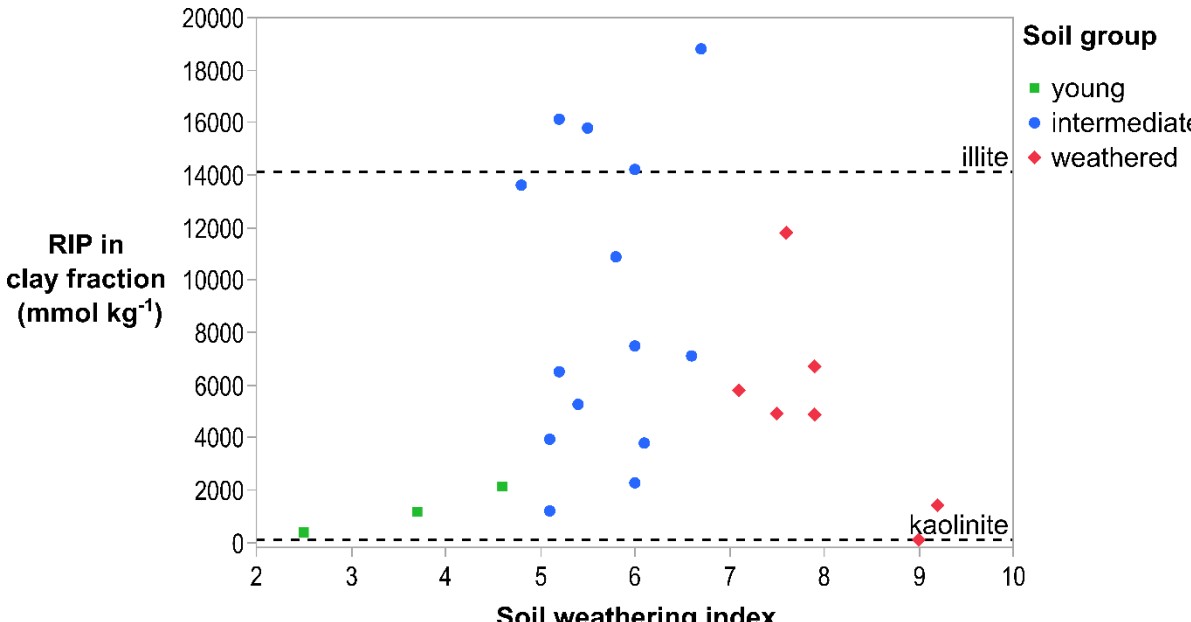

**Figure 2: RIPs in the clay fractions of soils as a function of soil weathering. The dashed lines are RIP of illite (GM = 14,100 mmol kg$^{-1}$; de Koning et al., 2007; Wauters et al., 1994) and kaolinite (GM = 77 mmol kg$^{-1}$; Nakao et al., 2008; Ogasawara et al., 2013). The data suggest an upward trend between young and intermediate weathering followed by a decrease in highly weathered soils.**





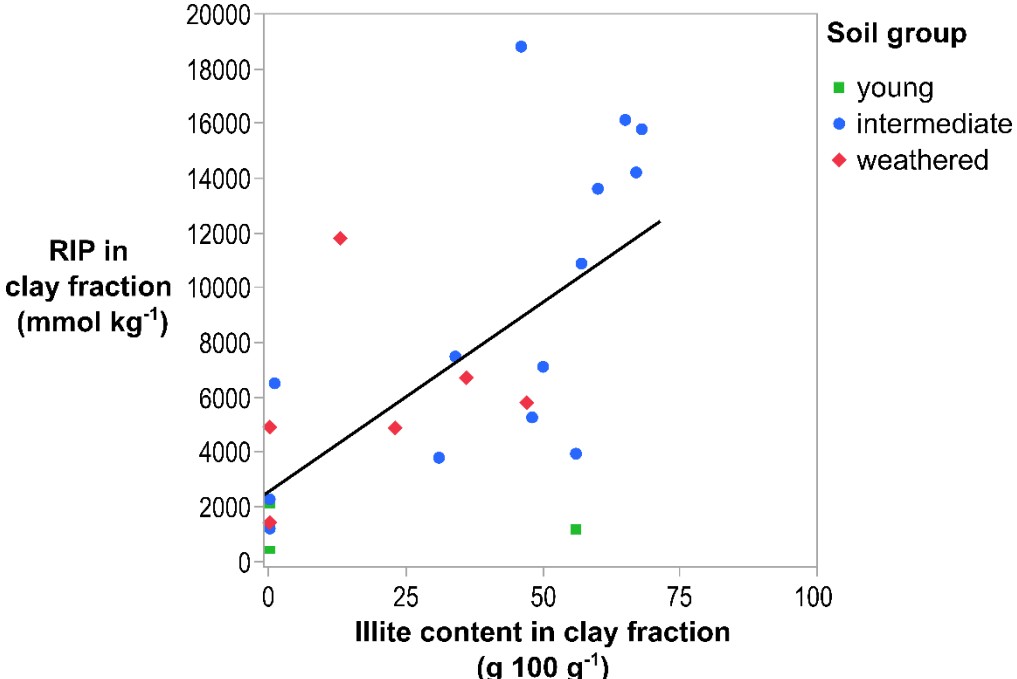

**Figure 3: The RIP in the clay fraction increases with increasing illite content of the clay fraction ($R^2 = 0.43$; *p* <0.001). The solid line is the regression line.**

As stated before, the bulk soil RIP is *unrelated* to the clay content in the soil (Fig. 1) and the clay fraction RIP is related to the illite content in the clay fraction (Fig. 3). Altogether, the bulk soil RIP is significantly related to the illite content in soil, i.e., the fraction illite in the clay fraction multiplied with the clay content ($R^2 = 0.50$; *p* <0.001; Graphical Abstract). Note that the illite content might be overestimated in some soils, as not all mineral phases in the clay fraction could be fully quantified.

For three soils, however, over 50% and up to 67% of the soil RIP was found in the silt fraction. The XRD detected weathered mica and other phyllosilicates in that texture fraction but these were not quantitatively converted to illite content because non-phyllosilicates, e.g., quartz and feldspars, would have led to inaccurate results (Fig. C27). The fraction of theoretical RIPs located in the silt fractions rises with the silt content, i.e., mass fraction of 2–50 µm soil particles ($R^2 = 0.18$; *p* <0.05; Fig. 4). The RIP in the bulk soil can be calculated from the RIP measured in the clay and the silt fractions multiplied by their corresponding mass fractions. These theoretical, i.e., calculated, values correlate well with RIPs independently measured in the bulk soil ($R^2 = 0.69$; *p* < 0.001; Fig. 5). However, the measured RIP in bulk soil was overestimated by this theoretical RIP by a factor of 2.0 on average and up to 9.1 for the soil from Pagsanjan (Philippines 5), suggesting that soil fractionation opens FES not accessible in the bulk soil (see discussion). The fraction of the theoretical RIP found in the clay fraction was logically high, averaging 76%.





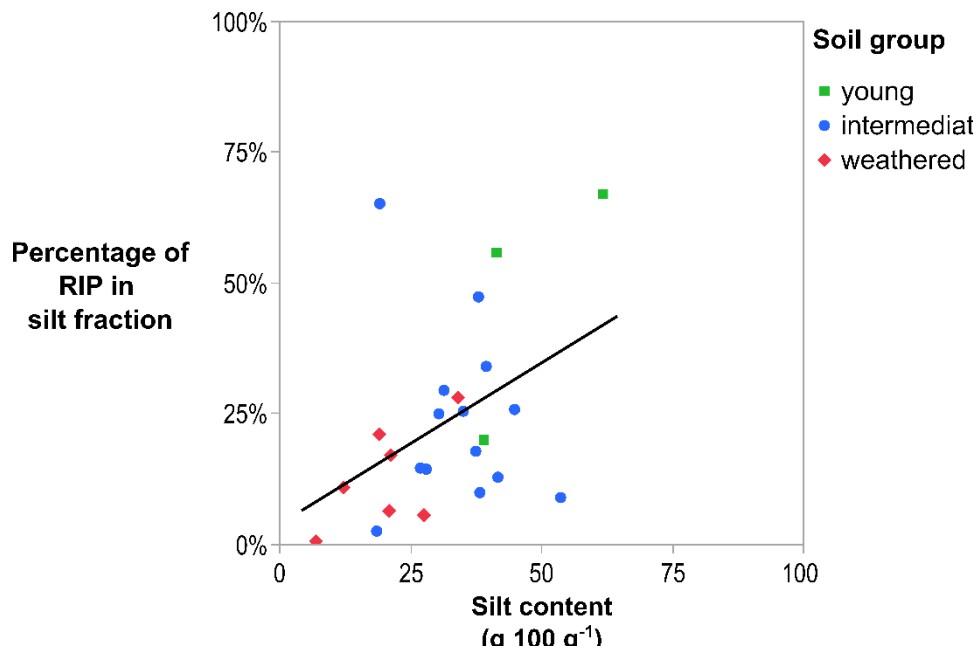


**Figure 4: The percentage of soil RIP present in the silt fraction increases with increasing silt content of the soil ($R^2 = 0.18$; $p < 0.05$). The solid line is the regression line.**



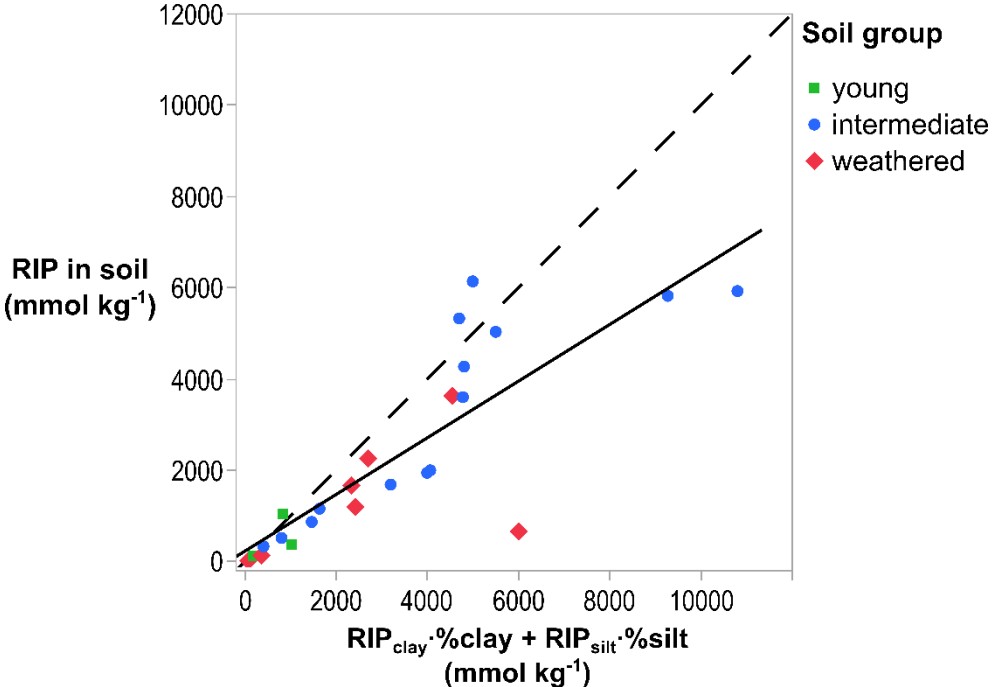

**Figure 5: Comparison of RIPs measured in bulk soils with the theoretical RIP calculated from the RIP in the clay and silt fractions multiplied with their corresponding mass fractions in soil ($R^2 = 0.69$; $p < 0.001$). The solid line is the regression line and the dashed line is the 1:1 line, illustrating, on average, a lower RIP based in the bulk soil than in the size fractions from that soil.**

## 4 Discussion

The Absalom model (Absalom et al., 2001; Tarsitano et al., 2011), which predicts radiocaesium bioavailability and is used for impact assessments (Brown et al., 2020), relies on the assumption that $^{137}$Cs sorption in soils is controlled by their clay content (0–2 µm). This implies that $^{137}$Cs sorbs mainly to minerals in the clay fraction and that the mineralogy has relatively small effects. This assumption may be acceptable in regions with similar mineralogy, but on a global scale, this premise is incorrect, as shown unequivocally in this study. First of all, the clay content does not control the RIP in bulk soil (Fig. 1) and, secondly, the mineralogy of that fraction has large effects on the RIP (Fig. 3). This study confirms (Vanheukelom et al., 2024) that $^{137}$Cs sorption in the soil is influenced by the mineralogy, which is determined by the parent material and the weathering stage. Soils with intermediate weathering stages, where illite dominates, show the highest RIP (>10,000 mmol kg$^{-1}$) in the clay fraction due to selective adsorption sites characteristic of illite. In contrast, the lowest RIP (<500 mmol kg$^{-1}$) is observed in the clay fraction of young soils, where selective adsorption sites in mica have not yet formed, and in highly weathered soils, where phyllosilicates with selective adsorption sites have disappeared (Fig. 2; Table 2). Soils dominated by illite show the highest RIP in the clay fraction, as illite contributes mainly to the RIP (Fig. 3). Compared to illite, other phyllosilicates, such as expanding phyllosilicates and kaolinite, contribute minimally to the RIP.



On average, 24% of the RIP in soils is found in the silt fraction. However, the contribution of the silt fraction to the total RIP increases as the silt content of the soil increases, suggesting that the silt fraction plays a notable role in $^{137}$Cs retention in soils with higher silt content. The XRD patterns in this study show phyllosilicates are mainly present in the clay fraction. However, weathered mica and kaolinite peaks are also observed in the silt fraction, questioning the reliability of the clay cut-off for

phyllosilicates. A previous study on Japanese soils found that RIP in the clay fraction, multiplied by the clay content (RIP$_{clay}$ %clay), underestimated the RIP in the bulk soil samples (RIP$_{soil}$) by a factor 2 (up to factor 5) (Uematsu et al., 2015). This discrepancy was most notable in soils with lower clay content, such as Cambisols and Gleysols, and was attributed to $^{137}$Cs-sorbing minerals in larger soil fractions. Alternatively, not all clay particles were recovered from the soils given that a subsample was taken of the clay fraction with a pipet (20 mL subsample of 500 mL clay+silt suspension). Another study on

Japanese soils, in which the clay fraction was isolated by siphoning (the entire suspension), found that RIP$_{clay}$ %clay did not match RIP$_{soil}$ (Nakao et al., 2015). For soils where RIP$_{clay}$ %clay underestimated RIP$_{soil}$, it was suggested that mica in larger fractions, derived from aeolian deposits from Chinese deserts, is a major contributor to the RIP. Conversely, for soils where the RIP$_{clay}$ %clay overestimated RIP$_{soil}$, it was suggested that selective adsorption sites in the isolated clay fraction were made accessible or reactive, which would not be exposed in the bulk soil due to interactions with organic matter.

This study examined the clay and silt fractions to quantify mineralogy and $^{137}$Cs adsorption. The Na$^+$-resin may have attracted K from the clay mineral interlayers causing more FES to form. In addition, we mechanically broke up soil aggregates to release phyllosilicate particles and did not chemically remove organic matter or other soil cementing materials to represent soil conditions. The sum of the $^{137}$Cs adsorption in both fractions overestimates RIP$_{soil}$ by a factor of 2.0, likely related to creating and opening up FES not accessible in bulk soils, even when relying on the more gentle resin method for soil fractionation.

This non-additivity of reactions in both fractions shows the limitation of soil fractionation to study the soil chemical reactions in the bulk soils. In parallel, non-additive behavior has also been observed for other heavy metals adsorbing to soil components like metal oxides and organic matter due to interactions between organic matter and oxide on the metal cation adsorption (Christl and Kretzschmar, 2001; Vermeer et al., 1999).

In the long term, the accessibility of radiocaesium to selective adsorption sites in soil aggregates and its migration into deeper

mineral structures may contribute to the effect of aging, with the radiocaesium solid-liquid distribution (and RIP) in soil increasing over time. Such suggests, but does not prove, that the theoretical RIP observed after soil fractionation may still be the correct one in the long term. Previous research concluded that adsorption increases in expanding phyllosilicates such as montmorillonite (Degryse et al., 2004; Maes et al., 1985), biotite (Vanheukelom et al., 2023), and in soils containing these phyllosilicates (Roig et al., 2007; Uematsu, 2017) after inducing collapses of mineral open edges on which radiocaesium is

selectively adsorbed (McKinley et al., 2004). In addition, radiocaesium would migrate into deeper mineral structures over time, as observed in illite (Fuller et al., 2015) common in European soils and in weathered biotite (Mukai et al., 2016) common in Fukushima soils. Soil aggregates and soil structure may also contribute to the increasing adsorption of radiocaesium over time, highlighting the need to understand the fate of radiocaesium in soils under field conditions over longer time frames.



This study falsified the assumptions of the Absalom model and its variants to predict the adsorption of radiocaesium in soils,
they are clearly not valid on a global scale. However, it was shown that these models could still predict the *bioavailability* of
radiocaesium with reasonable accuracy even in soils not initially included in the calibration of these models, e.g., in highly
weathered soils, due to counteracting effects. That is because, on the one hand, they overestimate RIPs in weathered soils, but
on the other hand, they underestimate K availability in the same soils (Vanheukelom et al., 2024). Hence, despite incorrect
assumptions, these models are still useful after recalibration using site-specific soil properties when available, until new and
better models are developed. These new models may be based on soil mineralogy, but it is already clear from soil maps that
information on soil mineralogy is not available globally; hence, proxies for this will be needed for implementing better models
in global soil information systems.

## 5 Conclusion

This study reveals that $^{137}$Cs sorption in soils is not controlled by the clay content (0–2 μm) but rather by the mineralogy,
which is influenced by parent material and weathering stages. Among phyllosilicates, illite contributes most to the $^{137}$Cs
sorption occurring mainly in the smallest (<2 μm) soil fractions. However, this was not true for soils with weathered mica in
the larger (2–50 μm) soil fractions. The $^{137}$Cs sorption in soil fractions overestimated that in bulk soil by a factor of 2, probably
due to the breaking up of soil structure and aggregates, thus artificially increasing the accessibility of adsorption sites for
radiocaesium in the short term. This calls for caution when extrapolating the fate of radiocaesium in isolated fractions from
laboratory studies to structured soils under field conditions.

## 6 Data availability

Additional information is given on the validation of the Na$^+$-resin method (Appendix A), the soil bulk mineralogy and
characteristics (Appendix B), the interpretation of XRD patterns and clay mineralogy (Appendix C), and correlation
coefficients (Appendix D). The X-ray diffraction patterns are openly available (Vanheukelom, Margot (2024),
"Vanheukelom2025_XRD-pattern", Mendeley Data, V1, doi: 10.17632/nr4f6s23k6.1).

## 7 Author contribution

The manuscript was written through contributions of all authors. All authors have given approval to the final version of the
manuscript. **Margot Vanheukelom**: Conceptualization, Data curation, Formal analysis, Investigation, Methodology,
Validation, Visualization, Writing – original draft, Writing – review & editing. **Nina Haenen**: Data curation, Formal analysis,
Methodology. **Talal Almahayni**: Funding acquisition, Project administration, Resources, Supervision, Writing – review &
editing. **Lieve Sweeck**: Funding acquisition, Project administration, Resources, Supervision. **Nancy Weyns**: Data curation,



Formal analysis, Methodology. **May Van Hees**: Data curation, Formal analysis, Methodology. **Erik Smolders**: Conceptualization, Formal analysis, Funding acquisition, Investigation, Methodology, Project administration, Resources, Supervision, Visualization, Writing – original draft, Writing – review & editing.

**8 Competing interests**

The authors declare that they have no conflict of interest.

**9 Acknowledgements**

The authors would like to thank Bruno Delvaux, Anne Iserentant, and Elodie Devos from Soil Science Laboratory (UCLouvain) for sharing their expertise and protocol on the dispersion and isolation of fractions from various soils. The authors
state that they did not use generative AI assistance tools during the research/writing process of this manuscript, except for mere language assistance. The authors have reviewed and edited the content as needed and take full responsibility for the content of the publication.

**10 Financial support**

SCK CEN is thanked for the PhD grant that was used to support the research of the manuscript.



## 11 Appendices

### 11.1 Appendix A

The method to isolate 0–2 µm and 2–50 µm fractions using Na[+]-resin (Rouiller et al., 1972) was validated by comparing it to a standardized pipet method for particle size analysis (ISO, 1998) with reference soils. The internal reference soil from the Division of Soil and Water Management (KU Leuven, Belgium) was sampled in Leuven, Belgium (50°52'42"N, 4°39'24"E), was used as arable land (rotation wheat and peas) and was classified as Haplic Luvisol (IUSS Working Group WRB FAO, 2022). The soil was air-dried and milled to pass a 2 mm sieve. The interlaboratory reference soils from Wageningen Evaluating Programmes for Analytical Laboratories (WEPAL) of the International Soil-analytical Exchange program included clayey soil from Ivory Coast (reference 883), sandy soil from Droevendaal, the Netherlands (reference 997), and organic Ferralsol from Sumatra Barat, Indonesia (reference 998). The soils were oven-dried (40 °C), milled to pass a 0.5 mm sieve, and sampled using an automatic sample divider. Each reference sample had mean values and statistics of soil characteristics (2010 and 2012 reports).

Particle distribution analyzed by resin method included mechanical disaggregation of soil particles by sonication, dispersion by a Na[+]-resin, and isolation of the 0–2 µm fraction by siphoning. In contrast, the ISO 11277 method included chemical disaggregation of soil particles by removing organic matter and salts (optional removal of iron oxides and carbonates), dispersion by a hexametaphosphate, and sampling the 0–2 µm fraction by pipet. In short, following the ISO 11277 method, 10–20 g of air-dry soil was milled to pass a 2 mm sieve. The soils were suspended in deionized water (250 mL) and treated with hydrogen peroxide (15%) to remove organic matter, followed by removing soluble salts with deionized water. Between each step, suspended soils were centrifuged (15 min; 2000 $g$), and clear supernatant was removed by decanting. The dispersing agent was made by dissolving 33 g of sodium hexametaphosphate (HMP) and 7 g of anhydrous sodium carbonate in 1 L of water. The sand fractions were separated by wet sieving (63 µm sieve). The remaining fractions were put in cylinders (500 mL) in a water bath (30°C), in which they equilibrated overnight (16 h). The contents of the cylinders were agitated using a plunger at a speed of 30 times per minute for 2 min. According to Stoke's law, after 6 h 9 min 45 sec, a sample (20 mL) was taken with a pipet 10 cm below the surface. All subsamples were oven-dried (105 °C; minimum two days), cooled to room temperature in a desiccator, and weighed analytically (accuracy 0.0001 g; OHAUS Adventurer™ AR2140). Corrections were made for moisture (105 °C) and the weight of HMP salt to calculate the weight of each fraction. Most results fell within an acceptable range of variation (Table A1), indicating that the resin method was reliable for particle size distributions. A representative 0–2 µm fraction was successfully isolated in this study using this method.



**Table A1: Validation of the particle size analysis method following ISO 11277 and procedure with a Na⁺-resin (used in this study to isolate the 0–2 µm and 2–50 µm fractions) by comparing results of reference soil samples. Number of measurements are given (*N*), mean values, and standard deviations.**

| location | | ISO 11277 method | | | resin method | | |
|---|---|---|---|---|---|---|---|
| | *N* | <2 µm | *N* | >63 µm | <2 µm | >50 µm | recovery |
| city, country | | g 100 g$^{-1}$ | | g 100 g$^{-1}$ | g 100 g$^{-1}$ | g 100 g$^{-1}$ | % |
| Leuven, Belgium | 29 | 7 ± 2 | 4 | 13 ± 3 | 15 | 13 | 100 |
| Ivory Coast | 30 | 9 ± 1 | 18 | 77 ± 2 | 8 | 78 | 99 |
| Droevendaal, Netherlands | 125 | 4 ± 1 | 70 | 80 ± 5 | 4 | 82 | 99 |
| Sumatra Barat, Indonesia | 35 | 83 ± 8 | 19 | 5 ± 1 | 82 | 8 | 94 |





## 11.2 Appendix B

**Table B1: Mineralogy of the bulk soils previously determined (Vanheukelom et al., 2024). Soils are ordered by increasing weathering index (WI) and grouped as young (WI = 2.5–4.7), intermediate (WI = 4.7–7.0), or weathered (WI = 7.0–9.2). qtz = quartz; mica = mica minerals; Al-2:1 = aluminium 2:1 phyllosilicates; Fe-2:1 = iron 2:1 phyllosilicates; Fe-2:1:1 = iron chlorite phyllosilicates; Mg-2:1:1 = magnesium chlorite phyllosilicates; 1:1 = kaolin phyllosilicates; fsp = K-feldspars and plagioclase; cal/dol = calcite and dolomite; po = pyrrhotite; Ant = anatase; FeOx = iron (oxy-)hydroxides; AlOx = aluminium (oxy-)hydroxides; amp/px = amphibole and pyroxene; a = mineral phases that appear amorphous on XRD patterns.**


| location | | | | | | mineralogy bulk soil | | | | | | | | | | | WI |
|---|---|---|---|---|---|---|---|---|---|---|---|---|---|---|---|---|---|
| | qtz | mica | Al-2:1 | Fe-2:1 | Fe-2:1:1 | Mg-2:1:1 | 1:1 | fsp | cal/dol | po | ant | FeOx | AlOx | amp/px | a | |
| city, country | | | | | | | g 100 g$^{-1}$ | | | | | | | | | | - |
| mineral weathering index: | *6* | *7* | *7* | *4* | *4* | *8* | *9* | *5* | *2* | *1* | *12* | *11* | *10* | *3* | *1* | |
| *young* | | | | | | | | | | | | | | | | |
| Miyakonojo, Japan | 4 | 7 | <1 | <1 | <1 | <1 | <1 | 22 | <1 | <1 | <1 | <1 | <1 | <1 | 67 | 2.5 |
| Mount Elgon, Kenya (1) | 3 | 6 | 6 | <1 | <1 | 7 | 9 | 7 | <1 | <1 | <1 | 3 | 3 | <1 | 56 | 3.7 |
| Pagsanjan, Philippines (1) | 1 | 11 | <1 | <1 | <1 | <1 | 15 | 37 | <1 | 1 | <1 | <1 | <1 | 4 | 31 | 4.6 |
| *intermediate* | | | | | | | | | | | | | | | | |
| Sidney, United States | 35 | 1 | 14 | <1 | <1 | <1 | 1 | 24 | 1 | <1 | <1 | <1 | <1 | 1 | 23 | 4.8 |
| Mount Elgon, Kenya (2) | 6 | 12 | 10 | <1 | <1 | <1 | 6 | 8 | <1 | 1 | 13 | 2 | <1 | <1 | 42 | 5.1 |
| Pagsanjan, Philippines (2) | 3 | 12 | <1 | <1 | <1 | <1 | 15 | 45 | <1 | 1 | <1 | <1 | <1 | 2 | 22 | 5.1 |
| Gross Enzersdorf, Austria | 38 | 12 | 6 | <1 | 5 | 3 | 2 | 11 | 22 | <1 | <1 | <1 | <1 | 1 | <1 | 5.2 |
| Ogata, Japan | 20 | <1 | 33 | <1 | <1 | <1 | 3 | 25 | <1 | <1 | <1 | <1 | <1 | <1 | 19 | 5.2 |
| Mount Elgon, Kenya (3) | 8 | 7 | 13 | <1 | <1 | <1 | 13 | 5 | <1 | <1 | 2 | 11 | 1 | <1 | 40 | 5.4 |
| Rots, France | 58 | <1 | 13 | <1 | 4 | <1 | <1 | 15 | 10 | <1 | <1 | <1 | <1 | <1 | <1 | 5.5 |
| Changchun, China | 36 | <1 | 17 | <1 | <1 | 6 | 4 | 35 | 2 | <1 | <1 | <1 | <1 | <1 | <1 | 5.8 |
| Pagsanjan, Philippines (3) | 4 | 16 | <1 | <1 | <1 | <1 | 33 | 21 | <1 | 1 | <1 | <1 | <1 | <1 | 25 | 6.0 |
| Isla Mayor, Spain | 16 | 7 | 43 | <1 | <1 | 6 | 2 | 11 | 15 | <1 | <1 | <1 | <1 | <1 | <1 | 6.0 |
| Yoshiki, Japan | 26 | 19 | 5 | <1 | <1 | 9 | 11 | 29 | <1 | <1 | <1 | <1 | <1 | 1 | <1 | 6.0 |
| Kitale, Kenya (1) | 61 | 3 | 4 | <1 | <1 | <1 | 10 | 14 | <1 | <1 | 1 | 1 | <1 | <1 | 6 | 6.1 |
| Mount Elgon, Kenya (4) | 10 | 6 | 16 | <1 | <1 | <1 | 18 | 8 | <1 | <1 | 3 | 15 | 2 | <1 | 22 | 6.6 |
| Aliminusa, Italy | 51 | <1 | 25 | <1 | <1 | <1 | 17 | 6 | <1 | <1 | 1 | <1 | <1 | <1 | <1 | 6.7 |
| *weathered* | | | | | | | | | | | | | | | | |
| Endebess, Kenya | 35 | 7 | 11 | <1 | <1 | <1 | 13 | 13 | <1 | <1 | 1 | 13 | 2 | <1 | 5 | 7.1 |
| Pagsanjan, Philippines (4) | 8 | 30 | <1 | <1 | <1 | <1 | 42 | 8 | <1 | 1 | <1 | 2 | <1 | <1 | 9 | 7.5 |
| Pagsanjan, Philippines (5) | 6 | 30 | <1 | <1 | <1 | <1 | 44 | 4 | <1 | 2 | <1 | 4 | <1 | <1 | 10 | 7.6 |
| Kitale, Kenya (2) | 28 | 7 | 13 | <1 | <1 | <1 | 24 | 5 | <1 | <1 | 2 | 15 | 2 | <1 | 4 | 7.9 |
| Kitale, Kenya (3) | 35 | 2 | 12 | <1 | <1 | <1 | 23 | 8 | <1 | <1 | <1 | 18 | 2 | <1 | <1 | 7.9 |
| Behenjy, Madagascar | 25 | <1 | <1 | <1 | <1 | 2 | 35 | <1 | <1 | <1 | 5 | 27 | 6 | <1 | <1 | 9.0 |
| Cavinti, Philippines | 3 | 12 | 5 | <1 | <1 | <1 | 68 | <1 | <1 | 1 | <1 | 11 | <1 | <1 | <1 | 9.2 |





**Table B2: Properties of the bulk soil previously determined (Vanheukelom et al., 2024). Soils are ordered by increasing WI and grouped as young (WI = 2.5–4.7), intermediate (WI = 4.7–7.0), or weathered (WI = 7.0–9.2). More details on soil origin can be found elsewhere (Vanheukelom et al., 2024).**

| | | particle size distribution | | | | | | | |
| location | WRB soil class | <2 μm | 2–50 μm | 50–2000 μm | organic C | pH | CEC | RIP bulk soil | WI |
| --- | --- | --- | --- | --- | --- | --- | --- | --- | --- |
| city, country | | g 100 g$^{-1}$ | | | | | cmol$_c$ kg$^{-1}$ soil | mmol kg$^{-1}$ soil | |
| *young* | | | | | | | | | |
| Miyakonojo, Japan | Andosol | 20 | 42 | 38 | 5.6 | 4.8 | 5.1 | 120 | 2.5 |
| Mount Elgon, Kenya (1) | Andosol | 29 | 62 | 9 | 14 | 4.1 | 6.2 | 360 | 3.7 |
| Pagsanjan, Philippines (1) | Cambisol | 31 | 39 | 30 | 1.8 | 5.9 | 24 | 1,000 | 4.6 |
| *intermediate* | | | | | | | | | |
| Sidney, United States | Kastanozem | 27 | 45 | 28 | 1.4 | 6.4 | 20 | 6,100 | 4.8 |
| Mount Elgon, Kenya (2) | Nitisol | 54 | 39 | 7 | 8.4 | 5.4 | 45 | 1,700 | 5.1 |
| Pagsanjan, Philippines (2) | Cambisol | 18 | 38 | 44 | 2.3 | 5.3 | 22 | 330 | 5.1 |
| Gross Enzersdorf, Austria | Chernozem | 26 | 42 | 32 | 1.0 | 7.4 | 13 | 3,600 | 5.2 |
| Ogata, Japan | Gleysol | 44 | 31 | 25 | 1.8 | 5.1 | 32 | 2,000 | 5.2 |
| Mount Elgon, Kenya (3) | Nitisol | 65 | 28 | 7 | 7.9 | 6.3 | 41 | 1,900 | 5.4 |
| Rots, France | Luvisol | 27 | 54 | 19 | 1.3 | 7.4 | 20 | 5,300 | 5.5 |
| Changchun, China | Phaeozem | 36 | 38 | 26 | 1.4 | 7.1 | 33 | 4,300 | 5.8 |
| Pagsanjan, Philippines (3) | Cambisol | 48 | 31 | 21 | 1.8 | 5.6 | 34 | 860 | 6.0 |
| Isla Mayor, Spain | Vertisol | 59 | 38 | 3 | 1.7 | 7.8 | 26 | 5,800 | 6.0 |
| Yoshiki, Japan | Cambisol | 8 | 19 | 73 | 0.83 | 5.2 | 9.8 | 1,100 | 6.0 |
| Kitale, Kenya (1) | Gleysol | 18 | 27 | 55 | 1.3 | 4.1 | 5.4 | 510 | 6.1 |
| Mount Elgon, Kenya (4) | Nitisol | 58 | 35 | 7 | 5.3 | 5.2 | 50 | 5,000 | 6.6 |
| Aliminusa, Italy | Cambisol | 56 | 19 | 25 | 0.81 | 5.8 | 26 | 5,900 | 6.7 |
| *weathered* | | | | | | | | | |
| Endebess, Kenya | Vertisol | 38 | 21 | 41 | 2.0 | 4.5 | 19 | 1,700 | 7.1 |
| Pagsanjan, Philippines (4) | Cambisol | 36 | 34 | 30 | 1.8 | 4.3 | 23 | 1,200 | 7.5 |
| Pagsanjan, Philippines (5) | Cambisol | 48 | 28 | 24 | 1.8 | 4.8 | 28 | 660 | 7.6 |
| Kitale, Kenya (2) | Ferralsol | 56 | 21 | 23 | 1.8 | 3.8 | 8.3 | 3,600 | 7.9 |
| Kitale, Kenya (3) | Ferralsol | 44 | 19 | 37 | 1.8 | 4.8 | 8.1 | 2,200 | 7.9 |
| Behenjy, Madagascar | Ferralsol | 68 | 12 | 20 | 2.9 | 4.2 | 2.3 | 14 | 9.0 |
| Cavinti, Philippines | Acrisol | 25 | 7 | 68 | 0.56 | 3.8 | 9.9 | 130 | 9.2 |





**11.3 Appendix C**

The identification of phyllosilicate mineral phases in X-ray diffraction (XRD) patterns of isolated 0–2 μm and 2–50 μm fractions from soils was performed following the procedure of Moore and Reynolds (1997). The patterns were shifted to position the quartz peak around 3.34 Å. A mineral phase was identified based on the peak position, corresponding to the largest peak in the XRD pattern. The smaller peaks were located to confirm its presence, and these peaks were subsequently discarded from consideration. This iterative process was continued by examining the remaining peaks to identify additional mineral

phases, repeating the steps until all observable peaks were attributed to specific minerals. Mineral phases not included in the Sybilla software library were identified using tables in Brindley and Brown (1980). However, some difficulties were encountered due to high background noise and overlapping peaks of different mineral phases. These interferences made it challenging to identify specific minerals, leaving some peaks unassigned, such as in Fig. C3.

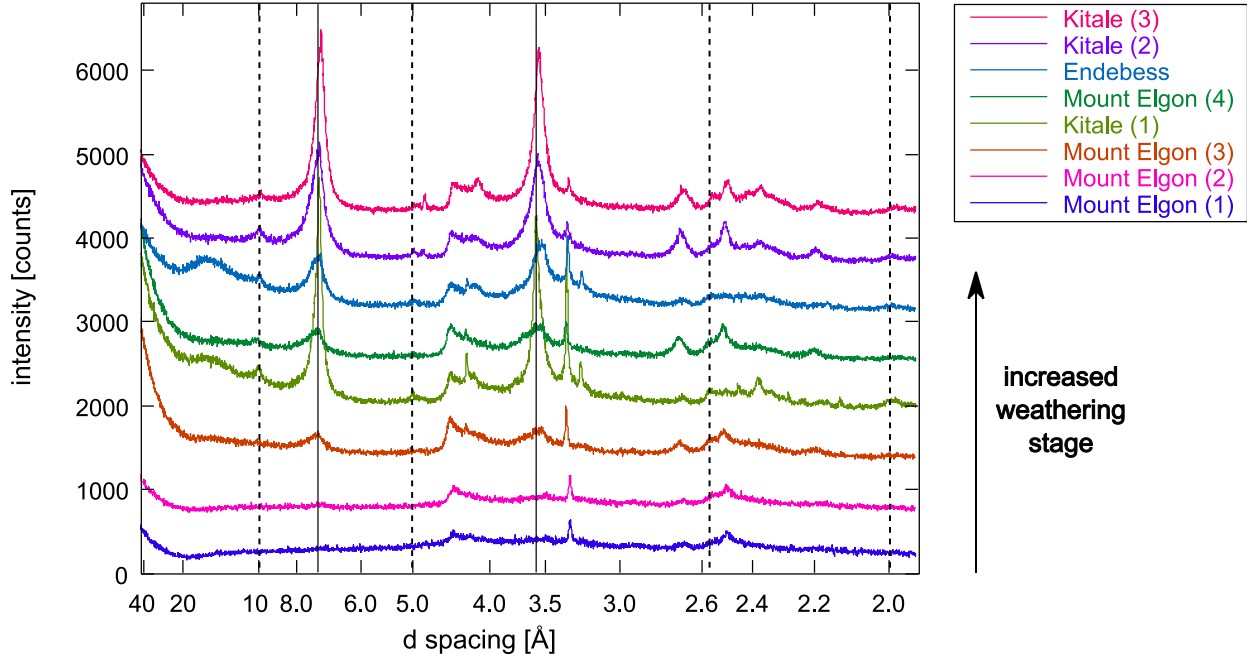

**Figure C1: Comparison of the oriented clay fractions in air-dried, Ca-saturated soils from Kenya. Full lines indicate peak positions of kaolinite at 7.17 Å and 3.58 Å, and dotted lines indicate peak positions of illitic mineral phases at 10 Å, 5.0 Å, 3.36 Å, 2.56 Å, and 2.00 Å.**



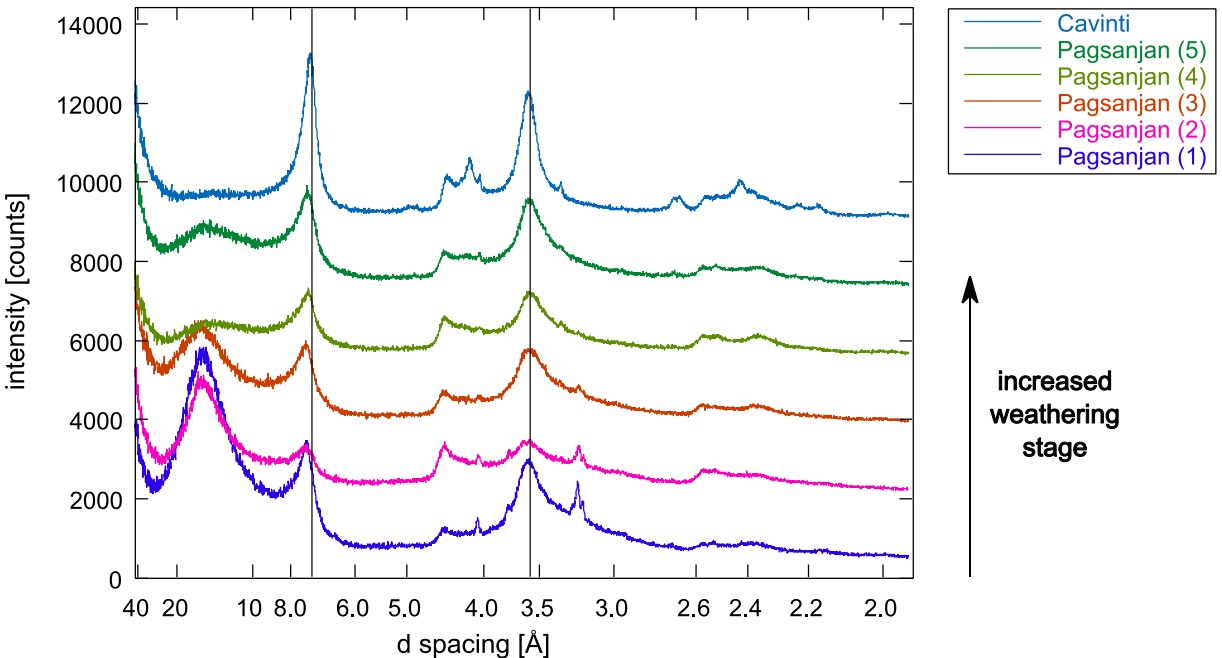

**Figure C2: Comparison of the oriented clay fractions in air-dried, Ca-saturated soils from the Philippines. The XRD patterns shown here were measured with variable slit but converted to fixed slit. Full lines indicate peak positions of kaolinite at 7.17 Å and 3.58 Å.**

The quantification of phyllosilicate mineral phases in XRD patterns of isolated 0–2 µm fractions from soils was performed following the procedure of Zeelmaekers (2011). Mineral phases were quantified using Sybilla software (© Chevron ETC) by fitting the model of operator-selected mineral models to the experimental XRD pattern in both air-dried Ca-form and ethylene glycol solvated states. The background was removed using a linear function. The patterns were shifted to position the kaolinite peak around 7.15 Å, which was clear and unaffected by ethylene glycol. To facilitate model calculations, the step size was increased from 0.02 °2θ to 0.05 °2θ, which was acceptable given the wide peaks of the phyllosilicates. Parts of the experimental XRD pattern that could not be modeled, such as quartz peaks or background at low angles (2–5 °2θ), were excluded from the fitting process. In the software library, kaolinite and chlorite were available as discrete mineral phases, while illite and smectite were fitted as mixed-layered mineral phases. Mixed-layered mineral phases with expandable layers, such as illite-smectite (high-charge), were fitted with two types of smectic layers: regular and high-charge. The high-charge layers with smaller d-spacings compared to regular smectite were used to model vermiculite-like swelling behavior. The proportions of the layers varied between 0 and 100%. For example, in Fig. C5, randomly interstratified kaolinite-smectite (high charge) with proportions 50:30:20 had a 50% kaolinite layer, 30% regular smectite, and 20% high-charge smectite. In this study, regular and high-charge smectite contents were summed together as expandable mineral phase. The quantified mineral phase results were accepted if the model fit for the ethylene glycol state agreed with the model for the air-dry state. The mineral phase contents were calculated as averages of the results of the two states.



Calcite was observed in many soils (Fig. C6; Fig. C9; Fig. C10; Fig. C12; Fig. C13; Fig. C15; Fig. C19; and Fig. C25) except the sequence soils. In the bulk soils, calcite was observed by XRD analysis (Table B1), which varied between <1–22 g 100 g$^{-1}$. In addition, inorganic carbon was previously measured in these soils, which varied between <0.10–3.0% inorganic carbon

(Vanheukelom et al., 2024). However, this was not expected for the more weathered soil of Behenjy (Fig. C25), in which inorganic carbon was not measured. Moreover, the pH (4.2) was lower than pKa (9.0), so calcite in the soil would probably have dissolved. It is assumed that the calcite in the soil is a precipitation product due to significant pH changes during the experimental procedure. The use of Na$^+$-resin (>500 µm; Amberlite® IR-120 (H); Supelco®) facilitated the exchange of H$^+$ ions for Na$^+$ ions, leading to a substantial increase in pH. When the soil was exposed to 1 mol L$^{-1}$ CaCl$_2$, the increased pH

promoted the formation of carbonate ions (CO$_3^{2-}$) from bicarbonate (HCO$_3^-$). The pH was 8.3 for Behenjy (Madagascar), measured in 0.2 g of 0–2 µm fraction in 2 mL purified water (Hanna HI 2002-02). This high concentration of Ca$^{2+}$ and CO$_3^{2-}$ ions resulted in the precipitation of CaCO$_3$ as calcite. Thus, the calcite peaks observed in XRD are probably the result of these experimental conditions rather than the presence of calcite in the original soil sample. No calcite was observed in the soil sequences, probably because the soil buffered the pH more. In the soil from Kitale (Kenya 1), the pH of the 0–2 µm fraction

was 7.6.



**Figure C3: The oriented clay fraction of soil from Miyakonojo (Japan). Top: experimental XRD patterns in air-dried, Ca-saturated state (black), and no pattern was modeled because no phyllosilicate could be identified; Bottom: experimental XRD patterns in ethylene glycol solvated, Ca-saturated state (black) and no pattern was modeled because no phyllosilicate could be identified. The arrows indicate quartz (qtz) and feldspar (fsp) peaks. Allophane shows broad peaks at 3.50 Å (25.45 °2θ) and 2.50 Å (35.90 °2θ).**





**Figure C4: The oriented clay fraction of soil from Mount Elgon (Kenya 1). Top: experimental XRD patterns in air-dried, Ca-saturated state (black) and the modeled pattern (red) of 30% kaolinite, 45% randomly interstratified illite-smectite (97:3), and 25% randomly interstratified illite-smectite (60:40); Bottom: experimental XRD patterns in ethylene glycol solvated, Ca-saturated state (black) and the modeled pattern (red) of 27% kaolinite, 24% randomly interstratified illite-smectite (97:3), and 49% randomly interstratified illite-smectite (60:40). The arrows indicate quartz (qtz), maghemite (mgh), and hematite (hem) peaks. Allophane shows broad peaks at 3.52 Å (25.29 °2θ) and 2.50 Å (35.85 °2θ).**






**Figure C5: The oriented clay fraction of soil from Pagsanjan (Philippines 1). Top: experimental XRD patterns in air-dried, Ca-
saturated state (black) and the modeled pattern (red) of 30% kaolinite and 70% randomly interstratified kaolinite-smectite (high
charge) (50:30:20); Bottom: experimental XRD patterns in ethylene glycol solvated, Ca-saturated state (black) and the modeled
pattern (red) of 30% kaolinite, and 70% randomly interstratified kaolinite-smectite (high charge) (50:30:20). The arrows indicate
feldspar (fsp) peaks.**







**Figure C6: The oriented clay fraction of soil from Sidney (United States). Top: experimental XRD patterns in air-dried, Ca-saturated state (black) and the modeled pattern (red) of 6% kaolinite, 16% randomly interstratified illite-smectite (97:3), 59% randomly interstratified illite-smectite (high charge) (72:14:14), and 19% randomly interstratified illite-smectite (20:80); Bottom: experimental XRD patterns in ethylene glycol solvated, Ca-saturated state (black) and the modeled pattern (red) of 8% kaolinite, 6% randomly interstratified illite-smectite (97:3), 66% randomly interstratified illite-smectite (high charge) (72:14:14), and 20% randomly interstratified illite-smectite (20:80). The arrows indicate quartz (qtz), and calcite (cal) peaks.**



**Figure C7: The oriented clay fraction of soil from Mount Elgon (Kenya 2). Top: experimental XRD patterns in air-dried, Ca-saturated state (black) and the modeled pattern (red) of 31% kaolinite, 45% randomly interstratified illite-smectite (97:3), and 24% randomly interstratified illite-smectite (60:40); Bottom: experimental XRD patterns in ethylene glycol solvated, Ca-saturated state (black) and the modeled pattern (red) of 26% kaolinite, 28% randomly interstratified illite-smectite (97:3), and 46% randomly interstratified illite-smectite (60:40). The arrows indicate quartz (qtz), anatase (ant), maghemite (mgh), and hematite (hem) peaks.**




**Figure C8: The oriented clay fraction of soil from Pagsanjan (Philippines 2). Top: experimental XRD patterns in air-dried, Ca-saturated state (black) and the modeled pattern (red) of 16% kaolinite and 84% randomly interstratified kaolinite-smectite (high charge) (50:30:20); Bottom: experimental XRD patterns in ethylene glycol solvated, Ca-saturated state (black) and the modeled pattern (red) of 16% kaolinite, and 84% randomly interstratified kaolinite-smectite (high charge) (50:30:20). The arrows indicate feldspar (fsp) peaks.**





**Figure C9: The oriented clay fraction of soil from Gross Enzersdorf (Austria). Top: experimental XRD patterns in air-dried, Ca-**
**saturated state (black) and the modeled pattern (red) of 3% kaolinite, 38% randomly interstratified illite-smectite (97:3), 48%**
**randomly interstratified illite-smectite (68:32), and 11% randomly interstratified chlorite-smectite (90:10); Bottom: experimental**
**XRD patterns in ethylene glycol solvated, Ca-saturated state (black) and the modeled pattern (red) 3% kaolinite, 27% randomly**
**interstratified illite-smectite (97:3), 50% randomly interstratified illite-smectite (68:32), and 20% randomly interstratified chlorite-**
**smectite (90:10). The arrows indicate quartz (qtz), feldspar (fsp), and calcite (cal) peaks.**








**Figure C10: The oriented clay fraction of soil from Ogata (Japan). Top: experimental XRD patterns in air-dried, Ca-saturated state (black) and the modeled pattern (red) of 15% kaolinite, 46% randomly interstratified kaolinite-smectite (high charge) (25:37:38), 39% randomly interstratified kaolinite-smectite (regular) (25:75), and 1% randomly interstratified illite-smectite (15:85); Bottom: experimental XRD patterns in ethylene glycol solvated, Ca-saturated state (black) and the modeled pattern (red) of 8% kaolinite, 65% randomly interstratified kaolinite-smectite (high charge) (25:37:38), 19% randomly interstratified kaolinite-smectite (regular) (25:75), and 8% randomly interstratified illite-smectite (15:85). The arrows indicate quartz (qtz), feldspar (fsp), and calcite (cal) peaks.**





**Figure C11: The oriented clay fraction of soil from Mount Elgon (Kenya 3). Top: experimental XRD patterns in air-dried, Ca-saturated state (black) and the modeled pattern (red) of 29% kaolinite, 29% randomly interstratified illite-smectite (97:3), and 42% randomly interstratified illite-smectite (60:40); Bottom: experimental XRD patterns in ethylene glycol solvated, Ca-saturated state (black) and the modeled pattern (red) of 39% kaolinite, 17% randomly interstratified illite-smectite (97:3), and 44% randomly interstratified illite-smectite (60:40). The arrows indicate quartz (qtz), anatase (ant), maghemite (mgh), and hematite (hem) peaks.**






**Figure C12: The oriented clay fraction of soil from Rots (France). Top: experimental XRD patterns in air-dried, Ca-saturated state (black) and the modeled pattern (red) of 9% kaolinite, 26% randomly interstratified illite-smectite (95:5), and 65% randomly interstratified illite-smectite (68:32); Bottom: experimental XRD patterns in ethylene glycol solvated, Ca-saturated state (black) and the modeled pattern (red) of 9% kaolinite, 19% randomly interstratified illite-smectite (95:5), and 72% randomly interstratified illite-smectite (68:32). The arrows indicate quartz (qtz), and calcite (cal) peaks.**






**Figure C13: The oriented clay fraction of soil from Changchun (China). Top: experimental XRD patterns in air-dried, Ca-saturated state (black) and the modeled pattern (red) of 9% kaolinite, 11% randomly interstratified kaolinite-smectite (regular) (50:50), 11% randomly interstratified illite-smectite (high charge) (33:33:34), 60% randomly interstratified illite-smectite (97:3), and 9% randomly interstratified illite-smectite (63:37); Bottom: experimental XRD patterns in ethylene glycol solvated, Ca-saturated state**
**(black) and the modeled pattern (red) of 6% kaolinite, 24% randomly interstratified kaolinite-smectite (regular) (50:50), 17% randomly interstratified illite-smectite (high charge) (33:33:34), 22% randomly interstratified illite-smectite (97:3), and 31% randomly interstratified illite-smectite (63:37). The arrows indicate quartz (qtz), feldspar peak (fsp), and calcite (cal) peaks.**



**Figure C14: The oriented clay fraction of soil from Pagsanjan (Philippines 3). Top: experimental XRD patterns in air-dried, Ca-saturated state (black) and the modeled pattern (red) of 10% kaolinite, and 90% randomly interstratified kaolinite-smectite (high charge) (66:17:17); Bottom: experimental XRD patterns in ethylene glycol solvated, Ca-saturated state (black) and the modeled pattern (red) of 10% kaolinite, and 90% randomly interstratified kaolinite-smectite (high charge) (66:17:17). The arrows indicate feldspar (fsp) peaks.**





**Figure C15: The oriented clay fraction of soil from Isla Mayor (Spain). Top: experimental XRD patterns in air-dried, Ca-saturated state (black) and the modeled pattern (red) of 9% kaolinite, 54% randomly interstratified illite-smectite (97:3), 31% randomly interstratified illite-smectite (50:50), and 6% chlorite; Bottom: experimental XRD patterns in ethylene glycol solvated, Ca-saturated state (black) and the modeled pattern (red) of 10% kaolinite, 51% randomly interstratified illite-smectite (97:3), 33% randomly interstratified illite-smectite (50:50), and 6% chlorite. The arrows indicate quartz (qtz) and calcite (cal) peaks.**






**Figure C16: The oriented clay fraction of soil from Yoshiki (Japan). Top: experimental XRD patterns in air-dried, Ca-saturated state (black) and the modeled pattern (red) of 56% kaolinite, 6% randomly interstratified illite-smectite (high charge) (29:28:43), and 38% randomly interstratified illite-smectite (97:3); Bottom: experimental XRD patterns in ethylene glycol solvated, Ca-saturated state (black) and the modeled pattern (red) of 48% kaolinite, 31% randomly interstratified illite-smectite (high charge) (29:28:43), and 21% randomly interstratified illite-smectite (97:3). The arrows indicate quartz (qtz), feldspar (fsp), and peak (gbs) peaks.**




**Figure C17: The oriented clay fraction of soil from Kitale (Kenya 1). Top: experimental XRD patterns in air-dried, Ca-saturated state (black) and the modeled pattern (red) of 53% kaolinite, 19% randomly interstratified illite-smectite (97:3), and 28% randomly interstratified illite-smectite (60:40); Bottom: experimental XRD patterns in ethylene glycol solvated, Ca-saturated state (black) and the modeled pattern (red) of 65% kaolinite, 16% randomly interstratified illite-smectite (97:3), and 19% randomly interstratified illite-smectite (60:40). The arrows indicate quartz (qtz), rutile (rt), and hematite (hem) peaks.**






**Figure C18: The oriented clay fraction of soil from Mount Elgon (Kenya 4). Top: experimental XRD patterns in air-dried, Ca-**
**saturated state (black) and the modeled pattern (red) of 31% kaolinite, 32% randomly interstratified illite-smectite (97:3), and 37%**
**randomly interstratified illite-smectite (60:40); Bottom: experimental XRD patterns in ethylene glycol solvated, Ca-saturated state**
**(black) and the modeled pattern (red) of 32% kaolinite, 17% randomly interstratified illite-smectite (97:3), and 51% randomly**
**interstratified illite-smectite (60:40). The arrows indicate quartz (qtz), anatase (ant), maghemite (mgh), and hematite (hem) peaks.**



**Figure C19: The oriented clay fraction of soil from Aliminusa (Italy). Top: experimental XRD patterns in air-dried, Ca-saturated state (black) and the modeled pattern (red) of 28% kaolinite, 8% randomly interstratified illite-smectite (97:3), 61% randomly interstratified illite-smectite (68:32), and 3% smectite; Bottom: experimental XRD patterns in ethylene glycol solvated, Ca-saturated state (black) and the modeled pattern (red) of 28% kaolinite, 5% randomly interstratified illite-smectite (97:3), 55% randomly interstratified illite-smectite (68:32), and 12% smectite. The arrows indicate quartz (qtz) and calcite (cal) peaks.**






**Figure C20: The oriented clay fraction of soil from Endebess (Kenya). Top: experimental XRD patterns in air-dried, Ca-saturated state (black) and the modeled pattern (red) of 29% kaolinite, 25% randomly interstratified illite-smectite (97:3), and 46% randomly interstratified illite-smectite (60:40); Bottom: experimental XRD patterns in ethylene glycol solvated, Ca-saturated state (black) and the modeled pattern (red) of 38% kaolinite, 13% randomly interstratified illite-smectite (97:3), and 49% randomly interstratified**
**illite-smectite (60:40). The arrows indicate quartz (qtz), anatase (ant), rutile (rt), and hematite (hem) peaks.**





**Figure C21: The oriented clay fraction of soil from Pagsanjan (Philippines 4). Top: experimental XRD patterns in air-dried, Ca-saturated state (black) and the modeled pattern (red) of 52% kaolinite and 48% randomly interstratified kaolinite-smectite (high charge) (33:33:34); Bottom: experimental XRD patterns in ethylene glycol solvated, Ca-saturated state (black) and the modeled pattern (red) of 41% kaolinite, and 59% randomly interstratified kaolinite-smectite (high charge) (33:33:34).**



**Figure C22: The oriented clay fraction of soil from Pagsanjan (Philippines 5). Top: experimental XRD patterns in air-dried, Ca-saturated state (black) and the modeled pattern (red) of 66% kaolinite, 2% randomly interstratified kaolinite-smectite (high charge) (60:24:16), and 32% randomly interstratified illite-smectite (high charge) (70:15:15); Bottom: experimental XRD patterns in ethylene glycol solvated, Ca-saturated state (black) and the modeled pattern (red) of 36% kaolinite, 58% randomly interstratified kaolinite-smectite (high charge) (60:24:16) and 6% randomly interstratified illite-smectite (high charge) (70:15:15).**





**Figure C23: The oriented clay fraction of soil from Kitale (Kenya 2). Top: experimental XRD patterns in air-dried, Ca-saturated state (black) and the modeled pattern (red) of 53% kaolinite, 27% randomly interstratified illite-smectite (97:3), and 20% randomly interstratified illite-smectite (60:40); Bottom: experimental XRD patterns in ethylene glycol solvated, Ca-saturated state (black) and the modeled pattern (red) of 55% kaolinite, 16% randomly interstratified illite-smectite (97:3), and 29% randomly interstratified illite-smectite (60:40). The arrows indicate gibbsite (gbs), goethite (gth), and hematite (hem) peaks.**







**Figure C24: The oriented clay fraction of soil from Kitale (Kenya 3). Top: experimental XRD patterns in air-dried, Ca-saturated** 555 **state (black) and the modeled pattern (red) of 73% kaolinite, 20% randomly interstratified illite-smectite (97:3), and 7% randomly interstratified illite-smectite (60:40); Bottom: experimental XRD patterns in ethylene glycol solvated, Ca-saturated state (black) and the modeled pattern (red) of 68% kaolinite, 10% randomly interstratified illite-smectite (97:3), and 22% randomly interstratified illite-smectite (60:40). The arrows indicate gibbsite (gbs), goethite (gth), and hematite (hem) peaks.**



**Figure C25: The oriented clay fraction of soil from Behenjy (Madagascar). Top: experimental XRD patterns in air-dried, Ca-saturated state (black) and the modeled pattern (red) of 54% kaolinite and 46% chlorite; Bottom: experimental XRD patterns in ethylene glycol solvated, Ca-saturated state (black) and the modeled pattern (red) of 40% kaolinite, and 60% chlorite. The arrows indicate quartz (qtz), gibbsite (gbs), goethite (gth), anatase (ant), hematite (hem), and calcite (cal) peaks.**





**Figure C26: The oriented clay fraction of soil from Cavinti (Philippines). Top: experimental XRD patterns in air-dried, Ca-saturated state (black) and the modeled pattern (red) of 38% kaolinite, and 62% randomly interstratified kaolinite-smectite (high charge) (80:10:10); Bottom: experimental XRD patterns in ethylene glycol solvated, Ca-saturated state (black) and the modeled pattern (red) of 42% kaolinite, and 58% randomly interstratified kaolinite-smectite (high charge) (80:10:10). The arrows indicate quartz (qtz), gibbsite (gbs), goethite (gth), and hematite (hem) peaks.**



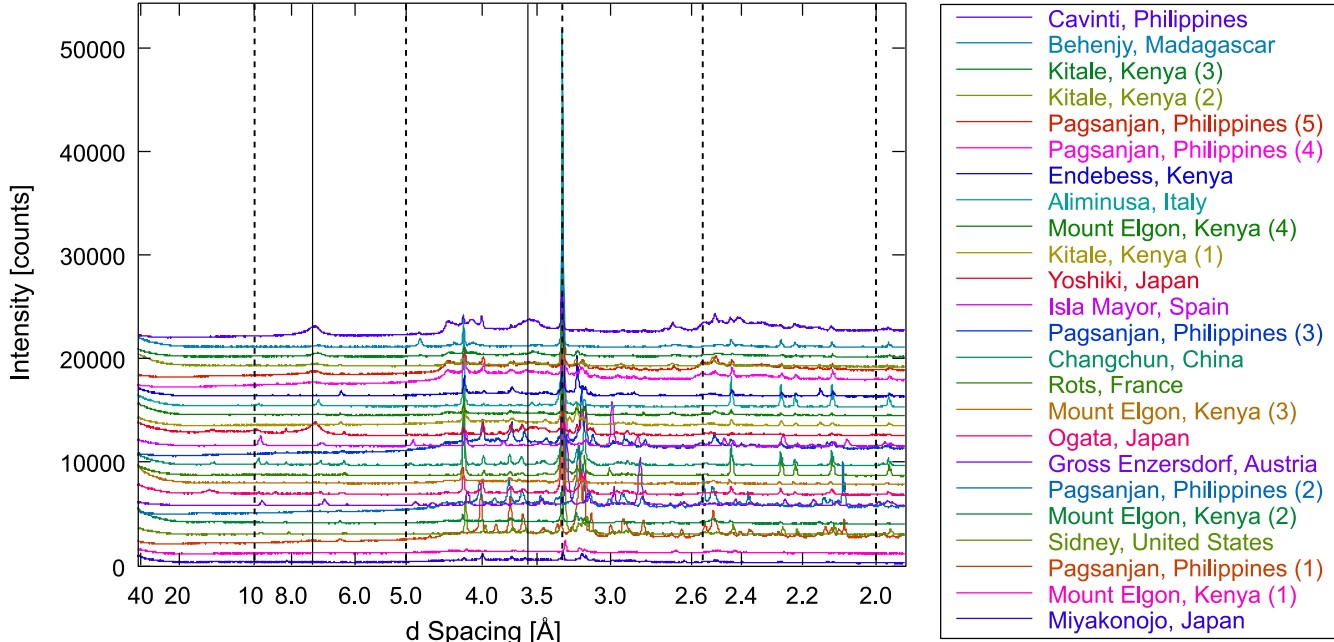


**Figure C27: Comparison of the oriented silt fractions in air-dried, Ca-saturated state of all soils in this study ordered from bottom to top by increased weathering stage. Full lines indicate peak positions of kaolinite at 7.17 Å and 3.58 Å, and dotted lines indicate peak positions of mica at 10 Å, 5.0 Å, 3.36 Å, 2.56 Å, and 2.00 Å. Quartz peaks are at 3.34 Å, 4.26 Å, and 2.46 Å.**



**Table C1: Mineralogy of the <2 μm fraction of soils. Soils are ordered by increasing WI and grouped as young (WI = 2.5–4.7), intermediate (WI = 4.7–7.0), or weathered (WI = 7.0–9.2). K = kaolinite; I = illite; S = regular smectite; $S^b$ = high-charge smectite, i.e., expandable layers with vermiculite-like swelling behavior; C = chlorite. Combinations are randomly mixed-layered (i.e., interstratified) mineral phases. Illite is modeled as mixed-layered illite-smectite with 3–5% smectite ($IS^c$) to match the illite peaks better. Ratios show the percentage of the mineral phase. Note: Mineral phases quantified are limited to illite, vermiculite, smectite, kaolinite, chlorite, and their mixed layers; other phases (e.g., allophane, mica, feldspars) were not included in the analysis.**

| location | K | KSS | ratio KSS | | | KS | ratio KS | | ISS | ratio ISS | | | $IS^c$ | ratio $IS^c$ | | IS | ratio IS | | SS | C | CS | ratio CS | |
| city, country | g 100 g⁻¹ | g 100 g⁻¹ | %K | %S | %$S^b$ | g 100 g⁻¹ | %K | %S | g 100 g⁻¹ | %I | %S | %$S^b$ | g 100 g⁻¹ | %I | %S | g 100 g⁻¹ | %I | %S | g 100 g⁻¹ | g 100 g⁻¹ | g 100 g⁻¹ | %C | %S |
| *young* | | | | | | | | | | | | | | | | | | | | | | | |
| Miyakonojo, Japan | <0.5 | <0.5 | | | | <0.5 | | | <0.5 | | | | <0.5 | | | <0.5 | | | <0.5 | <0.5 | <0.5 | | |
| Mount Elgon, Kenya (1) | 28.5 | <0.5 | | | | <0.5 | | | <0.5 | | | | 34.5 | 97 | 3 | 37.0 | 60 | 40 | <0.5 | <0.5 | <0.5 | | |
| Pagsanjan, Philippines (1) | 30.0 | 70.0 | 50 | 30 | 20 | <0.5 | | | <0.5 | | | | <0.5 | | | <0.5 | | | <0.5 | <0.5 | <0.5 | | |
| *intermediate* | | | | | | | | | | | | | | | | | | | | | | | |
| Sidney, United States | 7.0 | <0.5 | | | | <0.5 | | | 62.5 | 72 | 14 | 14 | 11.0 | 97 | 3 | 19.5 | 20 | 80 | <0.5 | <0.5 | <0.5 | | |
| Mount Elgon, Kenya (2) | 28.5 | <0.5 | | | | <0.5 | | | <0.5 | | | | 36.5 | 97 | 3 | 35.0 | 60 | 40 | <0.5 | <0.5 | <0.5 | | |
| Pagsanjan, Philippines (2) | 16.0 | 84.0 | 50 | 30 | 20 | <0.5 | | | <0.5 | | | | <0.5 | | | <0.5 | | | <0.5 | <0.5 | <0.5 | | |
| Gross Enzersdorf, Austria | 3.0 | <0.5 | | | | <0.5 | | | <0.5 | | | | 32.5 | 97 | 3 | 49.0 | 68 | 32 | <0.5 | <0.5 | 15.5 | 90 | 10 |
| Ogata, Japan | 11.5 | 55.5 | 25 | 37 | 38 | 29.0 | 25 | 75 | <0.5 | | | | <0.5 | | | 4.5 | 15 | 85 | <0.5 | <0.5 | <0.5 | | |
| Mount Elgon, Kenya (3) | 34.0 | <0.5 | | | | <0.5 | | | <0.5 | | | | 23.0 | 97 | 3 | 43.0 | 60 | 40 | <0.5 | <0.5 | <0.5 | | |



| location | K | KSS | ratio KSS | | | KS | ratio KS | | ISS | ratio ISS | | | IS[c] | ratio IS[c] | | IS | ratio IS | | SS | C | CS | ratio CS | |
|---|---|---|---|---|---|---|---|---|---|---|---|---|---|---|---|---|---|---|---|---|---|---|---|
| city, country | g 100 g$^{-1}$ | g 100 g$^{-1}$ | %K | %S | %S[b] | g 100 g$^{-1}$ | %K | %S | g 100 g$^{-1}$ | %I | %S | %S[b] | g 100 g$^{-1}$ | %I | %S | g 100 g$^{-1}$ | %I | %S | g 100 g$^{-1}$ | g 100 g$^{-1}$ | g 100 g$^{-1}$ | %C | %S |
| Rots, France | 9.0 | <0.5 | | | | <0.5 | | | <0.5 | | | | 22.5 | 95 | 5 | 68.5 | 68 | 32 | <0.5 | <0.5 | <0.5 | | |
| Changchun, China | 7.5 | <0.5 | | | | 17.5 | 50 | 50 | 14.0 | 33 | 33 | 34 | 41.0 | 97 | 3 | 20.0 | 63 | 37 | <0.5 | <0.5 | <0.5 | | |
| Pagsanjan, Philippines (3) | 10.0 | 90.0 | 66 | 17 | 17 | <0.5 | | | <0.5 | | | | <0.5 | | | <0.5 | | | <0.5 | <0.5 | <0.5 | | |
| Isla Mayor, Spain | 9.5 | <0.5 | | | | <0.5 | | | <0.5 | | | | 52.5 | 97 | 3 | 32.0 | 50 | 50 | <0.5 | 6.0 | <0.5 | | |
| Yoshiki, Japan | 52.0 | <0.5 | | | | <0.5 | | | 18.5 | 29 | 28 | 43 | 29.5 | 97 | 3 | <0.5 | | | <0.5 | <0.5 | <0.5 | | |
| Kitale, Kenya (1) | 59.0 | <0.5 | | | | <0.5 | | | <0.5 | | | | 17.5 | 97 | 3 | 23.5 | 60 | 40 | <0.5 | <0.5 | <0.5 | | |
| Mount Elgon, Kenya (4) | 31.5 | <0.5 | | | | <0.5 | | | <0.5 | | | | 24.5 | 97 | 3 | 44.0 | 60 | 40 | <0.5 | <0.5 | <0.5 | | |
| Aliminusa, Italy | 28.0 | <0.5 | | | | <0.5 | | | <0.5 | | | | 6.5 | 97 | 3 | 58.0 | 68 | 32 | 7.5 | <0.5 | <0.5 | | |
| *weathered* | | | | | | | | | | | | | | | | | | | | | | | |
| Endebess, Kenya | 33.5 | <0.5 | | | | <0.5 | | | <0.5 | | | | 19.0 | 97 | 3 | 47.5 | 60 | 40 | <0.5 | <0.5 | <0.5 | | |
| Pagsanjan, Philippines (4) | 46.5 | 53.5 | 33 | 33 | 34 | <0.5 | | | <0.5 | | | | <0.5 | | | <0.5 | | | <0.5 | <0.5 | <0.5 | | |
| Pagsanjan, Philippines (5) | 51.0 | 30.0 | 60 | 24 | 16 | <0.5 | | | 19.0 | 70 | 15 | 15 | <0.5 | | | <0.5 | | | <0.5 | <0.5 | <0.5 | | |
| Kitale, Kenya (2) | 54.0 | <0.5 | | | | <0.5 | | | <0.5 | | | | 21.5 | 97 | 3 | 24.5 | 60 | 40 | <0.5 | <0.5 | <0.5 | | |



| location | K | KSS | ratio KSS | | | KS | ratio KS | | ISS | ratio ISS | | | IS$^c$ | ratio IS$^c$ | | IS | ratio IS | | SS | C | CS | ratio CS | |
|---|---|---|---|---|---|---|---|---|---|---|---|---|---|---|---|---|---|---|---|---|---|---|---|
| city, country | g 100 g$^{-1}$ | g 100 g$^{-1}$ | %K | %S | %S$^b$ | g 100 g$^{-1}$ | %K | %S | g 100 g$^{-1}$ | %I | %S | %S$^b$ | g 100 g$^{-1}$ | %I | %S | g 100 g$^{-1}$ | %I | %S | g 100 g$^{-1}$ | g 100 g$^{-1}$ | g 100 g$^{-1}$ | %C | %S |
| Kitale, Kenya (3) | 70.5 | <0.5 | | | | <0.5 | | | <0.5 | | | | 15.0 | 97 | 3 | 14.5 | 60 | 40 | <0.5 | <0.5 | <0.5 | | |
| Behenjy, Madagascar | 47.0 | <0.5 | | | | <0.5 | | | <0.5 | | | | <0.5 | | | <0.5 | | | <0.5 | 53.0 | <0.5 | | |
| Cavinti, Philippines | 40.0 | 60.0 | 80 | 10 | 10 | <0.5 | | | <0.5 | | | | <0.5 | | | <0.5 | | | <0.5 | <0.5 | <0.5 | | |




### 11.4 Appendix D

**Table D1: Pearson correlation coefficients estimated by pairwise method between measured variables ($N$ = 24) and the illite equivalent clay. Significant correlations are \*\*\* $p$ <0.001; \*\* $p$ <0.01; \* $p$ <0.05; and not significant (n.s.) if $p$ >0.05. I = illite; S = expanding phyllosilicate (regular and high-charge smectite); C = chlorite; K = kaolinite.**

|  | 0–2000 µm | | | | | | | 2–50 µm | <2 µm | | | | |
|---|---|---|---|---|---|---|---|---|---|---|---|---|---|
|  | RIP$_{soil}$[e] | <2 µm[d] | 2–50 µm[d] | 50–2000 µm[d] | $^{10}$log(%C organic)[e] | pH[e] | CEC[e] | RIP$_{silt}$[d] | RIP$_{clay}$[d] | %I[d] | %S[d] | %C[d] | %K[d] |
| **0–2000 µm** | | | | | | | | | | | | | |
| RIP$_{soil}$ | - | | | | | | | | | | | | |
| <2 µm | n.s. | - | | | | | | | | | | | |
| 2–50 µm | n.s. | n.s. | - | | | | | | | | | | |
| 50–2000 µm | n.s. | -0.73*** | -0.47* | - | | | | | | | | | |
| $^{10}$log(%C organic) | n.s. | n.s. | 0.45* | -0.61** | - | | | | | | | | |
| pH | 0.66*** | n.s. | 0.45* | n.s. | n.s. | - | | | | | | | |
| CEC | n.s. | 0.46* | n.s. | -0.55** | n.s. | n.s. | - | | | | | | |
| **2–50 µm** | | | | | | | | | | | | | |
| RIP$_{silt}$ | n.s. | n.s. | n.s. | n.s. | n.s. | n.s. | n.s. | - | | | | | |
| **0–2 µm** | | | | | | | | | | | | | |
| RIP$_{clay}$ | 0.83*** | n.s. | n.s. | n.s. | -0.45* | 0.67*** | n.s. | n.s. | - | | | | |
| %I | 0.72*** | n.s. | 0.42* | n.s. | n.s. | 0.57** | n.s. | n.s. | 0.65*** | - | | | |
| %S | n.s. | n.s. | n.s. | n.s. | n.s. | n.s. | 0.42* | n.s. | n.s. | n.s. | - | | |
| %C | n.s. | n.s. | n.s. | n.s. | n.s. | n.s. | n.s. | n.s. | n.s. | n.s. | n.s. | - | |
| %K | -0.58** | n.s. | -0.59** | 0.43* | n.s. | -0.63** | n.s. | n.s. | -0.51* | -0.67*** | n.s. | n.s. | - |
| %IEC | 0.72*** | n.s. | 0.42* | n.s. | n.s. | 0.57** | n.s. | n.s. | 0.65*** | 1.00*** | n.s. | n.s. | -0.67*** |

[d]$n$ = 1; [e]$n$ = 3





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
