# Peer review of "The clay mineralogy rather than the clay content determines radiocaesium adsorption in soils on a global scale"

_EGUsphere, 2024_

## Author Response (AR1)

**Rebuttal letter egusphere-2024-3585**

**RC1: Atsushi Nakao**

*I enjoy reading this manuscript. It is well-written and includes scientific novelty. I fully agree with your central concept that "$^{137}$Cs sorption is not controlled by the clay content but rather by the mineralogy, which is influenced by parent material and weathering states" (line 299-300 in Conclusion) . Figure 2 provides us a nice view of this idea. Although a more sampling set would be more increased data reliability, I can imagine the time and labor work required for fractionation and measurement of RIP and the other related physicochemical properties.*

**Comment 1**

*However, the reason why illite is more dominated at the intermediate weathering stage requires a more careful discussion. In my opinion, many of the soils currently distributed around the world (especially in Europe and North America) that are intermediate in development are happen to be rich in illite, but there may be many that are not. For example, soils developed mainly from loess deposition (e.g. Chernozem) are rich in illite, whereas those from mafic or ultramafic materials (e.g. serpentinite rock and glassy volcanic ash) have no illite even being developed intermediately. These differences are controlled by original mineralogy of the parent materials. I doubt if illite will be formed in the Miyakonojo soil (Figure C3) developed from glassy volcanic ash, even times fly to the long future.*

**Reply 1**

Comment accepted.

L259–260: Added "This is because selective adsorption sites in mica are not yet formed, or because mica is absent from the clay fraction.".

L263–267: Added "Intermediate-weathered soils in this study derived from loess or sedimentary deposits, such as soils from Europe (e.g., Gross Enzersdorf), north America (e.g., Sidney) and Asia (e.g., Changchun), are dominated by illite. However, intermediate-weathered

soils derived from parent materials such as mafic rocks or volcanic ash (e.g., Miyakonojo) may not contain illite at all, even at a more advances stage of weathering, highlighting the role of the parent material on the soil mineralogy.".

**Comment 2**

*Four possible processes are considered behind the "illite accumulation": 1) neoformation during pedogenesis, 2) selective concentration as the other clay minerals are dissolved, 3) downsizing of coarser (sandy or silty) illite to clay fraction through weathering, 4) translocation of exogenic illite from other environments. Which do you think of the major possible process to cause the dominance of illite at intermediate weathering stage?*

1) *Neoformation of illite or other mica phases is not probable through chemical weathering reaction in pedogenesis. Isomorphic substitution of 4-fold $Al^{3+}$ in Si-O tetrahedron, which is required to provide a series of permanent negative charges in the interlayer site, cannot dominantly occur at normal pressure and temperature in soil environment(Mackenzie et al., 1987; Marsh et al., 2024). Most of the 2:1 phyllosilicates with a considerable tetrahedrally isomorphic substitution are, therefore, considered to be crystallized at higher temperature and pressure than soil environments (e.g. magmatic melt and pressure solution during diagenesis).*

2) *Selective concentrations of illite are also not probable because dissolution of allophane or other poorly crystalline alminosilicates often associated with neoformation of the other (more resistant to weathering) clay minerals as smectite or kaolinite (e.g. Figure C2 showing XRD for soil clays in the Philippines).*

3) *If parent material contains illite, downsizing through pedogenesis is the most probable process to increase illite content in clay fraction.*

4) *If soil sampling sites are located at lower elevation close to illite-rich bedrock area or at the region where aeolian dusts are frequently deposited, translocation of exogenic illite could be a major process of the "illite accumulation".*

*I believe a more careful discussion of the above-mentioned issues with adding more citations will increase the value of your study.*

*Mackenzie, K.J.D., Brown, I.W.M., Cardile, C.M., Meinhold, R.H., 1987. The thermal reactions of muscovite studied by high-resolution solid-state 29-Si and 27-Al NMR. J. Mater. Sci. 22, 2645–2654.*

*Marsh, A.T.M., Brown, A.P., Freeman, H.M., Walkley, B., Pendlowski, H., Bernal, S.A., 2024. Determining aluminium co-ordination of kaolinitic clays before and after calcination with electron energy loss spectroscopy. Appl. Clay Sci. 255, 107402.*

**Reply 2**

Comment accepted.

L268–275: Added "Several processes can explain the accumulation of illite in the clay fraction at the intermediate weathering stage. Coarser illite particles in the sand or silt fraction can be reduced by weathering to finer particles in the clay fraction. Illite can also accumulate from other locations, such as erosion from illite-rich bedrock or deposition of illite-rich aeolian dust from deserts or other regions that contain fine illite particles (Nakao et al., 2021). Other processes such as neoformation of illite under typical soil conditions is unlikely because the high temperature and pressure conditions required for illite crystallization (Mackenzie et al., 1987; Marsh et al., 2024) are absent in surface environments. Also, selective concentration of illite by dissolution of other phyllosilicates is unlikely because these processes usually result in the formation of other stable phyllosilicates, such as kaolinite or smectite, rather than illite (e.g., Fig. C2).".

L646–647: Added "Mackenzie, K. J. D., Brown, I. W. M., Cardile, C. M., and Meinhold, R. H.: The thermal reactions of muscovite studied by high-resolution solid-state 29-Si and 27-Al NMR, J Mater Sci, 22, 2645–2654, https://doi.org/10.1007/BF01082158, 1987.".

L656–658: Added "Marsh, A. T. M., Brown, A. P., Freeman, H. M., Walkley, B., Pendlowski, H., and Bernal, S. A.: Determining aluminium coordination of kaolinitic clays before and after

calcination with electron energy loss spectroscopy, Appl Clay Sci, 255, 107402, https://doi.org/10.1016/j.clay.2024.107402, 2024".

**Comment 3**

*Additionally, careful definition of "illite" is recommended. If my understanding is correct, illite content was determined by XRD peak intensity of 1.0 nm d-spacing. Although it is not bad, this approach cannot discriminate muscovite or biotite from the "illite". RIP values are largely different between muscovite, biotite, and illite, and are more variable depending on their weathering stage (Kitayama et al., 2020). Especially, muscovite shows very low RIP. The significant but not very high R2 value in Figure 3 may partly due to the inclusion of muscovite in some soils (e.g. Mount Elgon, Kenya). I understand that it is technically difficult to separate illite from muscovite, so it is good to point out the possibility that some white mica is mixed in.*

*Kitayama, R., Yanai, J., Nakao, A., 2020. Ability of micaceous minerals to adsorb and desorb caesium ions: Effects of mineral type and degree of weathering. Eur. J. Soil Sci. 71, 641–653.*

**Reply 3**

Comment accepted.

L152–153: Added "The identification of *illite* was based on the 10 Å peak, which overlaps with the peaks of micas, such as muscovite and biotite, so no distinction could be made between them. However, illite is expected in the clay fraction and mica in the silt fraction, so the 10 Å peak in the clay fraction is probably mainly illite.".

L229–231: Changed to "Note that the illite content in some soils may be overestimated, as not all mineral phases in the clay fraction could be fully quantified, and the 10 Å peak used for identification may include other micas, such as muscovite, which have a lower RIP.".

**Comment 4**

*Result showing in Figure 5 is excellent. Most of the plots are close to 1:1 line, which is probably due to the fractionation without chemical decomposition of SOM. If you add the result with SOM decomposition (just a comment, no need to add in this study), you can show the inhibition*

*effect of SOM coverage on illite for $^{137}$Cs adsorption. Although it is an exceptional case, unexpectedly high RIP$_{clay}$ + RIP$_{silt}$ for Pagsanjan (Philippines) soil is curious. Opening up FES not accessible in bulk soil may be one reason, but it looks too high increments. Further inspection of mineralogical composition for this soil might be interesting.*

**Reply 4**

Comment accepted.

L169–170: Changed to "In one soil (Pagsanjan 5; Fig. C22), a small peak at 11.3 Å suggested the presence of illite mixed with expanded phyllosilicates, but it was barely detectable in the XRD pattern after ethylene glycol treatment.".

L211–213: Changed to "However, a weathered soil from the Philippines sequence (Pagsanjan 5) had a more than factor of 5 higher RIP (12,000 mmol kg$^{-1}$) compared to other soils from that sequence (geometric mean (GM) = 2,100 mmol kg$^{-1}$; $N$ = 5).".

L225–229: Added to "In the clay fraction of a weathered soil (Pagsanjan 5), the RIP (12,000 mmol kg$^{-1}$) was unexpectedly high and could not be explained by the illite content (13 g 100 g$^{-1}$; Fig. 3). A mixed-layered illite with expanded phyllosilicates was identified (Fig. C22), unlike other soils from the sequence. This outlier should be verified by remeasuring the RIP, which was based on a single isolated clay sample, and performing advanced mineralogical analyses to confirm the identity of the mineral phase at 11.3 Å.".

L238–240: Changed to "However, the measured RIP in bulk soil was overestimated by this theoretical RIP by a factor of 2.0 on average and up to 9.1 for the soil from Pagsanjan containing mixed-layered illite (Philippines 5; Fig. C22), suggesting that in general soil fractionation opens FES that are not accessible in the bulk soil (see discussion)."

**RC2: Anonymous Referee #2**

*The fate of radiocaesium in soils is probably one of the most documented subjects of study compared to the behavior of other radioelements. It is pleasant to note that despite these numerous studies, this knowledge can still be improved.*

*The novelty of the current work is clearly based on three aspects:*

1. *the collection of soil, taken from different continents, which made it possible to cover a wide range of soils characterized by a variable stage of weathering (which was quantified by an index),*

2. *the mineralogical characterization of the clay fraction, rarely carried out in other studies,*

3. *the consideration of the silt fraction and its role in the selective adsorption of radiocaesium (to my knowledge, never estimated before).*

**Comment 1**

*These advances represent an obvious added value of this work and effectively complement the ancient study of Vandebroek et al. (JER 2012) which had notably analyzed the correlation between soil texture (and other parameters) and the RIP of soils on a global scale (not cited in this work).*

*The paper is well structured and written with a rigorous presentation of methods and treatment of results, and an attractive and convincing interpretation. I have just a few questions and have suggested minor changes that could lead to interesting clarifications.*

**Reply 1**

Comment accepted.

L36–37: Changed to "However, the above-mentioned soil-to-plant transfer models typically assume a uniform mineral composition, which does not accurately reflect the diversity in soil mineralogy on a global scale (Vandebroek et al., 2012)".

L691–693: Added "Vandebroek, L., Van Hees, M., Delvaux, B., Spaargaren, O., and Thiry, Y.: Relevance of Radiocaesium Interception Potential (RIP) on a worldwide scale to assess soil vulnerability to 137Cs contamination, J Environ Radioact, 104, 87–93, https://doi.org/10.1016/j.jenvrad.2011.09.002, 2012.".

**Comment 2**

*Title : For clarification, I would add "... on a global scale."*

**Reply 2**

Comment accepted.

Title: Changed to "The clay mineralogy rather than the clay content determines radiocaesium adsorption in soils on a global scale".

**Comment 3**

*L134 : The formulation is confusing: « finer texture »; do you mean a higher clay content, or something else? What is the relationship with 32 g 100 g-1 which seems to refer to the silt+sand fraction?*

**Reply 3**

Comment accepted. There is confusion: "finer textures" refers to the clay fraction and in Table 1 the clay (<2 µm) and sand fractions (50–2000 µm) are given.

L134: Changed to "most weathered soils had a higher clay content (43 g 100 g$^{-1}$)".

Table 1: Numbers updated.

**Comment 4**

*L141 : We have to regret the few number of young soils (N=3) which could have been higher to consolidate the conclusions of the study.*

**Reply 4**

Comment accepted.

L206–207: Added "Although only a small number of young soils were included in this study (*N* = 3), their low RIP is consistent with expectations for soils at this stage of weathering.".

**Comment 5**

*L196-199 : « … in mica. … in pure illite. … in kaolinite » - Of which size ?*

**Reply 5**

Comment accepted. The reference RIP values found in literature were measured on different particle size fractions: <2 µm (Nakao et al., 2008); 10–20 µm (Eguchi et al., 2015); bulk (de Koning et al., 2007; De Preter, 1990; Ogasawara et al., 2013; Wauters et al., 1994) of the reference mineral. For consistency, only reference RIP values are considered that were measured in the <2 µm for illite and kaolinite (Fig. 2) and in the 10–20 µm for mica.

L198–201: Changed to "The youngest soils had a RIP corresponding to that measured in mica (300 mmol kg$^{-1}$ in 10–20 µm fraction; Eguchi et al., 2015). Intermediate-weathered soils had a highly variable RIP, and soils dominated by illite had a RIP similar to that measured in illite (11,800 mmol kg$^{-1}$ in <2 µm fraction; Nakao et al., 2008). Weathered soils had a low RIP when illite was absent, corresponding to the RIP measured in kaolinite (6 mmol kg$^{-1}$ in <2 µm fraction;.Nakao et al., 2008).".

**Comment 6**

*L215, Figure 2 : We suppose that reference value for illite and kaolinite are also for a clay size fraction ? Not obvious like it is mentioned in the legend.*

**Reply 6**

Comment accepted. As discussed in Comment 5, only reference RIP values measured in clay sized fractions will be considered.

Figure 2: Reference lines illite (11,800 mmol kg$^{-1}$) and kaolinite (6 mmol kg$^{-1}$) updated.

L217–218: Changed to "The dashed lines are RIP of illite (11,800 mmol kg$^{-1}$ in <2 µm fraction; Nakao et al., 2008) and kaolinite (6 mmol kg$^{-1}$ in <2 µm fraction; Nakao et al., 2008).".

**Comment 7**

*L276 : To avoid confusion between Cs and heavy metals, I would write: « … for other trace elements like x or y … »*

**Reply 7**

Comment accepted.

L296–298: Changed to "In parallel, non-additive behavior has also been observed for other trace elements, such as copper and cadmium, adsorbing to soil components, such as metal oxides and organic matter, due to interactions between oxide and organic matter on the metal cation adsorption (Christl and Kretzschmar, 2001; Vermeer et al., 1999).".

**Comment 8**

L280 : Aging or ageing ?

**Reply 8**

Point taken. "Aging" refers to the process by which radiocaesium bioavailability decreases over time after being introduced into the soil. This article is written in US English spelling, so "aging" was preferred over "ageing", which refer to the same concept.

**Comment 9**

L296 : « hence, proxies for this will be needed for implementing better models » - That formulation seems a little elusive without more explicit recommandations or suggestions. Why not explore the combination of soil type (as tested by Vandebroek et al;) normalized by geological, climate and topography index for example ? According to your experience, a proposal would be appropriate to guide future research.

**Reply 9**

Comment accepted.

L316–320: Changed to "Although soil classifications may seem a promising proxy for soil mineralogy, they have been shown to be a poor predictor of RIP in soils worldwide (as

demonstrated by Vandebroek et al., 2012). We recommend exploring a combination of alternative proxies for soil mineralogy, that incorporate factors such as parent material (e.g., geology) and weathering stage (e.g., influenced by climate and topography). Such proxies could provide a practical way to develop and integrate improved radioactive caesium transfer models into global soil information systems.".

**Comment 10**

L299 : I would precise : « This study reveals that, on a global scale, ... »

**Reply 10**

Comment accepted.

L321: Added "on a global scale".